# SQuBa: Speech Mamba Language Model with Querying-Attention for Efficient Summarization

## Abstract

Abstractive Speech Summarization (SSum) becomes increasingly difficult as the input speech length grows. To address this, we present SQuBa (Speech Querying Mamba Architecture), an end-to-end model designed explicitly for efficient speech summarization. SQuBa leverages a querying-attention Mamba projector to condense extended acoustic features into compact semantic tokens, which are subsequently summarized by the Mamba Large Language Model (LLM). The architecture's computational complexity scales linearly with input length, enabling efficient handling of longer inputs. A two-stage training framework, complemented by bootstrapped Direct Preference Optimization (DPO) fine-tuning, empowers SQuBa to generate concise and coherent summaries. Experimental results demonstrate that SQuBa delivers competitive performance while significantly improving inference speed, making it ideal for real-world applications such as podcast and meeting transcriptions.

## 1 Introduction

Abstractive Speech Summarization (SSum) (Murray et al., 2010; Shang et al., 2018) is a task to generate textual summaries from spoken content. Unlike Abstractive Text Summarization (TSum) (Neto et al., 2002), SSum faces the added complexity of dealing with the computational and performance limitations associated with processing long speech prompt. As speech length increases, the key challenge shifts to efficiently extracting and summarizing critical information while remaining within computational constraints.

Previous models for speech summarization address the task through either a cascaded approach (Zhang et al., 2021; Zhong et al., 2021; Palaskar et al., 2019) combining ASR and TSum models, or an end-to-end approach (Matsuura et al., 2023; Kang & Roy, 2024; Shang et al., 2024), as illustrated in Fig. 1. With the rise of Multimodal Large Language Models (MLLMs), recent studies have shown that end-to-end models outperform cascaded models by leveraging implicit acoustic features and minimizing error propagation (Matsuura et al., 2023; Shang et al., 2024). However, these models still face significant computational challenges, especially when processing long audio inputs.

These limitations stem from the architecture of the Transformer model (Vaswani et al., 2017), which serves as the foundation for many large language models (LLMs). Although the Transformer excels at capturing long-range dependencies and integrating multimodal information, its self-attention mechanism scales quadratically with input sequence length, leading to substantial computational and memory overhead when processing long inputs. While the Transformer's design allows for efficient parallel training and supports large-scale model development, recent efforts to extend its context window have not entirely resolved the computational challenges associated with handling lengthy sequences.

Recently, Mamba (Gu & Dao, 2023), a variant of structured state space model (SSM) (Gu et al., 2022a;b), emerged as an alternative to the Transformer architecture to address these bottlenecks. Mamba introduces input-dependent selective scanning, enabling the model to focus on the most relevant parts of a sequence. It also employs a hardware-aware algorithm for efficient parallel computation, enhancing performance and often matching or surpassing that of Transformers. This adaptability has led to applications across various domains, including image processing Zhu et al. (2024);

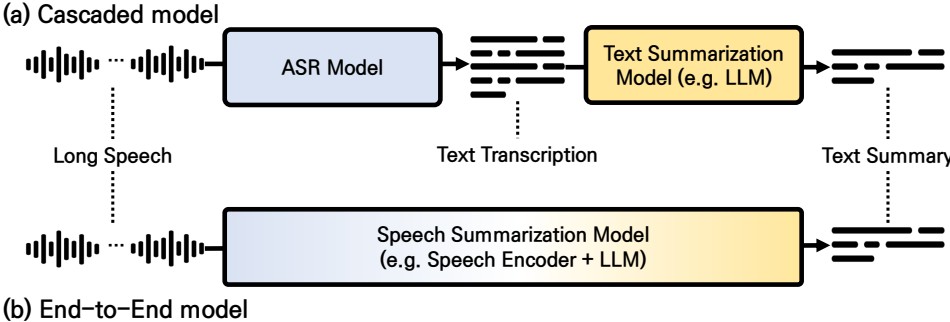

Figure 1: **Comparison between two types of speech summarization models: (a) cascaded model (top) and (b) non-cascaded end-to-end model (bottom).** The key distinction between these pipelines is whether an intermediate text transcription is generated. Our approach operates end-to-end, mitigating error propagation and inference delays while fully utilizing the acoustic information.

Liu et al. (2024), speech processing (e.g., speech separation) Jiang et al. (2024); Li & Guo (2024), video analysis Li et al. (2024), and multimodal LLMs (Qiao et al., 2024; Zhao et al., 2024).

To this end, we introduce SQuBa, an end-to-end Mamba-based speech summarization model designed to efficiently handle long speech prompt. SQuBa leverages a pre-trained Mamba LLM optimized for extended inputs, making it ideal for long speech prompt summarization tasks. To bridge the modality gap between speech and text, we propose the windowing Q-Mamba, inspired by the Q-Former (Li et al., 2023) used in recent Speech LLMs (Tang et al., 2024; Shang et al., 2024). The windowing Q-Mamba selectively compresses long speech prompt into compact latent semantic tokens via cross-attention and learnable queries, creating a computationally efficient architecture tailored for speech summarization. Through a two-stage training process and bootstrapped Direct Preference Optimization (Rafailov et al., 2023) fine-tuning, we demonstrate Mamba's ability to handle long speech prompt efficiently without compromising performance.

Our contributions are threefold:

- We introduce a query-attention Mamba projector, which compresses acoustic information from long speech prompts into compact semantic tokens, reducing the model's overall computational footprint.

- We extend the Mamba-based LLM to effectively handle lengthy speech inputs, demonstrating its speech summarization capabilities through our proposed SQuBa. We also present a two-stage training process for SQuBa, including bootstrapped DPO fine-tuning.

- We provide an empirical evaluation of SQuBa, highlighting its ability to achieve competitive performance with significantly lower computational demands, resulting in much faster inference speeds compared to transformer-based approaches.

## 2 RELATED WORKS

### 2.1 STATE SPACE MODEL (SSM) AND MAMBA

Classical state-space models (SSMs) represent 1-dimensional sequences using latent state matrices. The Linear State Space Layer (LSSL) (Gu et al., 2021), an early deep SSM, showed potential for modeling long-range dependencies but was limited by high computational costs. To improve efficiency, the Structured State-Space Model (S4) (Gu et al., 2022a) re-parameterized the latent matrix into low-rank and normal matrix components, leading to variants like DSS (Gupta et al., 2022) and S4D (Gu et al., 2022b), which optimized computation through diagonalization. However, S4 struggled with token retention and comparison, which are crucial for language modeling. Hungry Hungry Hippos (H3) (Fu et al., 2023) addressed this by incorporating a 1-dimensional convolution for token comparison and recall.

Mamba (Gu & Dao, 2023; Dao & Gu, 2024) builds on S4 with selective scanning and input-dependent latent parameters, allowing it to focus on relevant information. It incorporates 1-dimensional convolution from H3 and a gating mechanism similar to Long Short-Term Memory (Hochreiter & Schmidhuber, 1997), enhancing its handling of long sequences. With parallel scanning and hardware optimization, Mamba achieves efficient training and inference, often rivaling Transformers. It has also been adapted for applications in computer vision (Zhu et al., 2024; Liu et al., 2024; Patro & Agneeswaran, 2024) and speech (Li & Guo, 2024; Jiang et al., 2024).

## 2.2 Multimodal Large Language Models

Building on the success of Large Language Models (LLMs) (Ouyang et al., 2022; Touvron et al., 2023; Llama Team, 2024), Multimodal Large Language Models (MLLMs) extend LLMs to handle multimodal inputs, integrating various modalities beyond text. Notable models like LLaVA (Liu et al., 2023), BLIP (Li et al., 2022; 2023), and GPT-4 (OpenAI, 2024) use transformer architectures to manage long-range dependencies in multimodal data. In speech LLMs, SALMONN (Tang et al., 2024) has advanced the integration of auditory inputs–including speech, audio events, and music–via the windowing Querying Transformer (Q-Former). However, the high computational demands of these models have led to more efficient architectures like Cobra (Zhao et al., 2024) and VL-Mamba (Qiao et al., 2024), which enhance efficiency using the Mamba architecture (Gu & Dao, 2023) without sacrificing performance.

## 2.3 Speech Summarization

Speech summarization models fall into two categories: (1) cascaded models (Zhang et al., 2021; Zhong et al., 2021; Palaskar et al., 2019; 2021) and (2) end-to-end models (Sharma et al., 2021; Kano et al., 2023; Matsuura et al., 2023; Sharma et al., 2023; Shang et al., 2024). Cascaded models first transcribe speech into text using an automatic speech recognition (ASR) system, followed by a text summarization (TSum) model to generate summaries. In contrast, end-to-end models produce summaries directly from speech input, bypassing transcription. Although cascaded models initially benefited from pre-trained ASR and domain-specific TSum models, they suffer from longer inference times, error propagation due to transcription errors, and the inability to fully leverage acoustic information like intonation.

With advancements in multimodal language models, recent efforts have shifted toward end-to-end models, which have shown superior performance over cascaded approaches. However, due to the longer nature of speech inputs than text, Transformer-based end-to-end models face significant computational challenges as their complexity increases quadratically with input length. To address these issues, existing models employ input truncation (Matsuura et al., 2023), feature downsampling (Kang & Roy, 2024), or Q-former abstractor Shang et al. (2024). Our approach is similar to Shang et al. (2024) in its use of querying-attention for compact abstraction. Still, it differ by employing a more compact Mamba architecture, resulting in more efficient and faster training and inference.

## 3 Preliminaries

In this section, we introduce the preliminary concepts underlying our work. We start with an overview of State-Space Models (SSMs) and the Mamba architecture (Sec. 3.1). Next, we provide an overview of Direct Preference Optimization (DPO), which was used to align our model with the desired behavior (Sec. 3.2).

## 3.1 State-Space Models and Mamba

State-Space Models (SSMs) (Gu et al., 2021; 2022a) represent continuous systems that map input sequences $x(t)$ to output responses $y(t)$ via a hidden state $h(t)$. These models are typically characterized by a set of system parameters $(\mathbf{A}, \mathbf{B}, \mathbf{C}, \mathbf{D})$ that control the transformation of inputs into outputs:

$$h'(t) = \mathbf{A}h(t) + \mathbf{B}x(t)$$
$$y(t) = \mathbf{C}h(t) + \mathbf{D}x(t) \tag{1}$$

where the variable $\mathbf{D}$ is frequently viewed as a skip connection and is removed from the equation for simplicity. $h'(t)$ represents the time derivative of $h(t)$, or $dh(t)/dt$.

When dealing with discrete-time input sequences, these continuous models are discretized with matrices $\overline{\mathbf{A}}$ and $\overline{\mathbf{B}}$. A common method for discretization is the Zero-Order Hold (ZOH) technique, which computes these matrices as:

$$
\begin{aligned}
\overline{\mathbf{A}} &= \exp(\Delta\mathbf{A}) \\
\overline{\mathbf{B}} &= (\Delta\mathbf{A})^{-1} \left(\exp(\Delta\mathbf{A}) - \mathbf{I}\right) \Delta\mathbf{B}
\end{aligned}
\tag{2}
$$

where $\Delta$ represents the step size for discretization, and $\mathbf{I}$ is the identity matrix. The discretized state-space equations then become:

$$
\begin{aligned}
h_t &= \overline{\mathbf{A}}h_{t-1} + \overline{\mathbf{B}}x_t \\
y_t &= \mathbf{C}h_t
\end{aligned}
\tag{3}
$$

In Structured State-Space Model (S4) (Gu et al., 2022a), the parameters $(\mathbf{A}, \mathbf{B}, \mathbf{C}, \Delta)$ remain constant across all time steps, making the system time-invariant. While this simplifies the model, it limits its ability to adapt to varying inputs over time. To improve flexibility, Mamba (Gu & Dao, 2023) introduces input-dependent parameters for $\mathbf{B}$, $\mathbf{C}$, and $\Delta$, enabling a dynamic gating mechanism that selectively focuses on the most relevant information at each time step.

## 3.2 DIRECT PREFERENCE OPTIMIZATION

Direct Preference Optimization (DPO) (Rafailov et al., 2023) was introduced as a method for aligning LLMs to preference data without requiring additional training or the use of a reward model. Let $\pi_\theta$ be the LLM policy to be trained, parameterized with $\theta$, and $\pi_{\text{ref}}$ be the reference model, representing the initial (or supervised fine-tuned) state of the LLM. A preference dataset $\mathcal{D}_{\text{pref}} = \{(x, y_c, y_r)\}$ is given, where $x$ is an input prompt, $y_c$ is the preferred (chosen) response, and $y_r$ is the rejected response. The preference between $y_c$ and $y_r$ is relative, as only pairwise rankings of these prompt-response pairs are provided. The DPO loss is defined as:

$$
\mathcal{L}_{\text{DPO}}(\theta) = -\mathbb{E}_{(x,y_c,y_r)\sim\mathcal{D}}\left[\log\sigma\left(\beta\log\frac{\pi_\theta(y_c|x)}{\pi_{\text{ref}}(y_c|x)} - \beta\log\frac{\pi_\theta(y_r|x)}{\pi_{\text{ref}}(y_r|x)}\right)\right]
\tag{4}
$$

where $\sigma(\cdot)$ is a sigmoid function and $\beta$ is the hyperparameter that controls the divergence of the policy model $\pi_\theta$ from the reference model $\pi_{\text{ref}}(y|x)$. While this objective is derived from KL-constrained reward maximization, it can also be interpreted as maximizing the likelihood of the preferred response while minimizing the likelihood of the less preferred one.

## 4 SQuBA: SPEECH MAMBA LLM WITH QUERYING-ATTENTION

In this section, we introduce our query-based Mamba projector, explaining its adaptation for efficient speech feature extraction (Sec. 4.1). Then, we present the overall architecture of our speech summarization model, which integrates the speech encoder, query-based Mamba projector, and the Mamba LLM backbone (Sec. 4.2).

## 4.1 QUERYING-ATTENTION MAMBA (Q-MAMBA) PROJECTOR

We propose a querying-attention Mamba projector that leverages the Mamba layer, learnable queries, and cross-attention for efficient speech processing. The architecture of the projector is depicted on the right side of Figure 2. The core idea is to use a learnable query-based approach to compress speech features extracted by the Whisper encoder into a more compact sequence of tokens, retaining essential information while minimizing computational overhead. This allows for a much faster processing of long speech prompt prompt with minimal sacrificing of the quality of the extracted features.

The projector block consists of four key components: learnable queries, a unidirectional Mamba layer, cross-attention, and a feedforward network. Each query, initialized to cover approximately

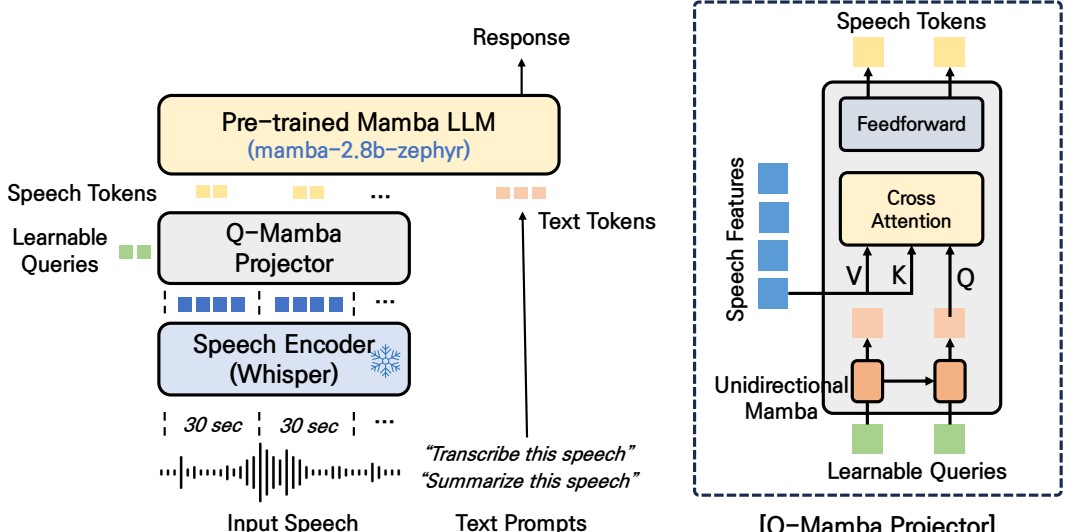

Figure 2: **Overall architecture of the proposed SQuBa (left) and the windowing querying-attention Mamba (Q-Mamba) projector (right).** The windowing Q-Mamba projects each speech feature chunk from the speech encoder into learnable queries via cross-attention. These projected speech features serve as speech token inputs for the pre-trained Mamba LLM backbone.

0.33 seconds of speech, forms causal dependencies through the Mamba layer. These queries then interact with the speech output from the pre-trained speech encoder, using cross-attention to selectively extract the most relevant information. The result is a compact, tokenized representation of the speech input, which is then passed to the Mamba-based LLM for further processing.

This approach allows for more efficient handling of long speech prompt by significantly reducing the length of the speech feature sequence, while dynamically focusing on relevant speech segments without overwhelming the model with extraneous data. Unlike vision-based cross-modal approaches that deal with static 2D image data, our speech projector is specifically designed to handle the temporal, variable-length nature of speech audio by windowing the projector across the speech feature sequence.

## 4.2 SPEECH SUMMARIZATION WITH MAMBA LANGUAGE MODEL

We present a speech summarization model built on our querying-attention Mamba projector. As shown in Figure 2, the architecture comprises a pre-trained speech encoder, our cross-modal projector, and a pre-trained Mamba LLM. First, the speech encoder extracts speech features from 30-second segments of input speech. These feature chunks are then processed by our projector, which generates queries embedded with projected semantic speech information. The output sequence is then combined with a tokenized text prompt and fed into the Mamba LLM to produce the corresponding text summarization.

## 5 TRAINING METHOD

In this section, we outline the two-stage training process for our proposed SQuBa model. This process is designed to: 1) effectively align the text and speech modalities, and 2) utilize the aligned speech representations to generate coherent and meaningful summaries. The overall process is depicted in Fig. 3.

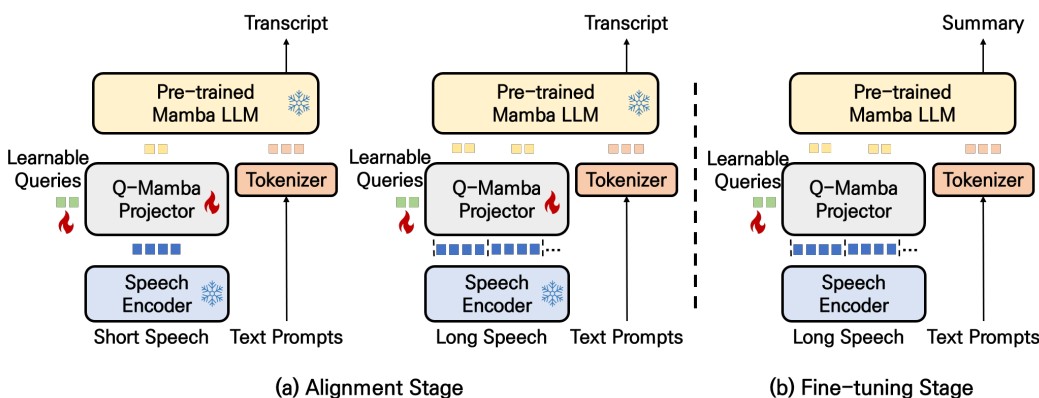

Figure 3: **Two-Stage Training Process of SQuBa.** In the alignment stage, only the projector is trained using an ASR task. In the fine-tuning stage, both the LLM backbone and the projector are trained on the summarization task. The first alignment step uses short speech inputs, while the second step and the fine-tuning stage process longer inputs that exceed the speech encoder's input length limit, creating chunks of speech tokens. Whisper encoder is frozen for all training stages.

## 5.1 SPEECH ALIGNMENT STAGE

In the alignment stage, the model is trained on an automatic speech recognition (ASR) task, with the parameters of the speech encoder and Mamba LLM backbone frozen. This allows the multimodal projector to effectively learn the mapping between speech and text modalities.

It's important to note that speech encoders typically have a maximum input length they can process. For instance, Whisper can only accept inputs in 30-second increments. To address this, speech longer than 30 seconds–referred to as **long-form speech**–is segmented into 30-second intervals, with each segment encoded separately. The model then concatenates the encoded features from the projector, ensuring the entire long speech prompt is processed coherently.

We employ a two-step approach to train the multimodal projector. First, the projector is trained on an easier dataset with shorter samples, enabling the model to focus on capturing core speech features and their corresponding textual representations without the complexity of long context dependencies. This initial step helps the model establish strong, localized alignments between speech and text before progressing to more complex, longer datasets.

In the second step, the model is trained on longer, more complex data that provides richer contextual information and a wider variety of speech patterns. Since the speech input exceeds the 30-second limit of the speech encoder, it is divided into feature chunks. By gradually increasing data complexity, the model becomes better equipped to handle extended speech sequences and align text and speech representations more effectively.

## 5.2 SUMMARIZATION FINE-TUNING STAGE

In this stage, the model is fine-tuned using supervised instruction-tuning to generate summaries from speech inputs of varying lengths. Speech sequences of different durations are provided, constrained only by the maximum capacity of the available GPU resources. This variation in input length helps the model generalize more effectively across a wide range of speech durations.

Similar to the second alignment stage, we process chunks of speech segments individually through the projector and then concatenate the features before passing them through the LLM. However, in this phase, we also unfreeze the pre-trained Mamba LLM and train it jointly with the projector, allowing for a more integrated optimization of the model.

**Bootstrapped DPO Fine-tuning** While the fine-tuned model generates text responses that are coherent with the speech content, we observed that the outputs often resembled transcriptions rather than summaries, resulting in undesirably long responses. To address this, we additionally apply Di-

rect Preference Optimization (DPO) (Rafailov et al., 2023) fine-tuning, a method that aligns LLMs with offline preference data without requiring an additional reward model.

To apply DPO to our model, we need a preference dataset $\mathcal{D}_{\text{pref}} = \{(x, y_c, y_r)\}$ where $x$ is the input (the speech and summarization prompt), $y_c$ is the preferred response, and $y_r$ is the rejected response. Since we only have the supervised fine-tuning dataset $\mathcal{D}_{\text{sup}} = \{(x, y)\}$, with $y$ as a ground-truth summarization, we use the outputs from the fine-tuned model as $y_r$ and the ground-truth as $y_c$. The initial state of the fine-tuned model serves as the reference model $\pi_{\text{ref}}$. To reduce training time, instead of generating summaries for each sample during training, we pre-generate them at the outset and use them as an off-policy dataset. We refer to this approach as bootstrapped DPO, as we bootstrap the process using our model's own responses.

Note that the Whisper encoder remains frozen for all training stages to leverage its pre-trained capabilities from large-scale multilingual audio data. This design choice reflects that the core challenge lies in cross-modal alignment between speech features and LLM embedding space, which is handled by the projector component. Training the projector alone maintains efficiency while avoiding encoder fine-tuning overhead, following established practices in multimodal LLM training (Liu et al., 2023; Zhou et al., 2024; Li et al., 2023; Chu et al., 2024) where pre-trained vision encoders remain frozen during cross-modal adaptation.

## 6 EXPERIMENTAL SETUP

**Models** For the pre-trained speech encoder, we use the encoder from Whisper Large v2 (Radford et al., 2023)[1]. For the pre-trained LLM backbone, we utilize the Mamba LLM (Gu & Dao, 2023), specifically Mamba-2.8B-zephyr[2], with 2.8 billion parameters, fine-tuned on UltraChat (Ding et al., 2023) and UltraFeedback (Cui et al., 2023). Detailed descriptions of each model can be found in Appendix A.1.

We use a query length of 2 for the querying-attention Mamba projector, which corresponds to approximately 0.33 seconds of speech. This compresses 0.33 seconds–into two queries, allowing the projector to capture semantic information as word-like speech tokens from the input. We justify this choice of query length in our ablation studies.

**Dataset** For the first step of the alignment stage, we use the Librispeech dataset (Panayotov et al., 2015), which contains 960 hours of English speech, with each sample under 30 seconds. In the second step, we utilize the XL subset of the Gigaspeech corpus (Chen et al., 2021), which includes 10,000 hours of English speech from diverse audio sources such as audiobooks and podcasts. For the main evaluation of the alignment stage, we use the test subset of Gigaspeech, while the test subset of Librispeech is used for ablation studies.

For the fine-tuning stage, we use a custom synthesized dataset based on the Mediasum dataset (Zhu et al., 2021), with synthetic speech generated via ChatTTS [3]. We limited the generated speech to 1500 tokens, which is about 6 minutes in length. See Appendix C.2 for more details. We additionally found that the ground-truth summaries from the original Mediasum dataset were found to be not ideal for training general-purpose LLMs. These summaries were often structured as news headlines or introductory sentences rather than complete summaries, making them inconsistent with how an LLM would naturally summarize content. This observation aligns with prior findings (Kang & Roy, 2024), which also noted that headline-based summaries are inadequate for training effective summarization models. To address this, we generated new ground-truth summaries using the pre-trained LLaMA 3 model (Llama Team, 2024) [4], with 8 billion parameters, to better align with the structure and format typically expected in LLM-based summarization.

**Metrics** To evaluate the performance of our speech summarization model, we use two widely recognized metrics for text summarization evaluation: ROUGE (Lin, 2004) and METEOR (Banerjee & Lavie, 2005). These metrics provide quantitative assessments of how closely the generated summaries align with reference summaries, considering factors such as content overlap and fluency.

---

[1] `https://huggingface.co/openai/whisper-large-v2`

[2] `https://huggingface.co/xiuyul/mamba-2.8b-zephyr`

[3] `https://github.com/2noise/ChatTTS`

[4] `https://huggingface.co/meta-llama/Meta-Llama-3-8B`

Table 1: **Summarization results on the Mediasum corpus with different input modalities.** **Mamba** refers to the Mamba-2.8B-Zephyr model fine-tuned on the Mediasum text corpus. **Cascaded** represents the cascaded system combining Whisper-Large-v2 and Mamba-2.8B-Zephyr. The third section compares end-to-end speech summarization models.

| Model | Input Modality | ROUGE-1 | ROUGE-2 | ROUGE-L | METEOR |
|---|---|---|---|---|---|
| Mamba | Text | 39.8 | 17.6 | 22.8 | 33.9 |
| Cascaded | Speech | 18.8 | 7.12 | 12.9 | 30.9 |
| Kang & Roy (2024) | Speech | 19.2 | 6.7 | 14.1 | 27.4 |
| SQuBa (ours) | Speech | **31.5** | **10.6** | **20.6** | **33.2** |

Table 2: **Comparison between different speech summarization models based on human evaluation.** Results show pairwise comparison (win rate %) between models across 5 human annotators, where a higher percentage indicates more preferred outputs. Each annotator evaluated 52 randomly sampled test examples from the MediaSum test set.

| Model vs. Model | win-rate (%:%) |
|---|---|
| SQuBa (ours) vs. Cascaded | **67.8** : 32.2 |
| SQuBa (ours) vs. Kang & Roy (2024) | **92.6** : 7.4 |
| Cascaded vs. Kang & Roy (2024) | **79.2** : 20.8 |

**Training Configurations** We trained the alignment stage for 2 epochs and extended training to 6 epochs during the fine-tuning stage to further refine the model's summarization capabilities. For the bootstrapped DPO stage, we ran 1 epoch to prevent overfitting. During the alignment stage, we used a learning rate of $10^3$, while a learning rate of $2 \times 10^5$ was used for the summarization fine-tuning and bootstrapped DPO. A cosine decay learning rate scheduler with a warmup ratio of 0.03 was applied throughout the training process.

For the first step of alignment stage, we employed 8 NVIDIA A6000 GPUs, each with 48GB of memory. In subsequent stages, we transitioned to 8 NVIDIA A100 GPUs, each with 80GB of memory, to accommodate the increased data complexity and model requirements. A global batch size of 128 was distributed across all GPUs.

## 7 EXPERIMENTAL RESULTS

### 7.1 MAIN RESULT

We compare our results against several baselines, including a cascaded version of our model and a recent end-to-end SSum model proposed by Kang & Roy (2024). The cascaded system uses the Whisper-Large-v2 model to transcribe speech inputs into text, which the fine-tuned Mamba-2.8B-zephyr model then summarizes. See Appendix C.1 for more cascaded model details. In contrast, Kang & Roy (2024) employs HuBERT and MiniChat2-3b, connected via Q-Former.

The results, shown in Table 1, demonstrate that SQuBa outperforms the baseline models. Specifically, SQuBa exhibits significant improvements over the cascaded Whisper and Mamba LLM system, as well as the model proposed by Kang & Roy (2024). We observed that transcription errors accumulated by Whisper significantly degrade the summarization quality of the cascaded model.

Human evaluation results in Table 2 further validate SQuBa's effectiveness, with annotators strongly preferring our summaries over baselines. SQuBa wins 67.8% and 92.6% of comparisons against the cascaded baseline and Kang & Roy (2024), respectively, confirming its ability to produce more coherent summaries. The cascaded baseline's 79.2% win rate over Kang & Roy (2024) suggests LLM-based approaches generally produce higher quality summaries. These results demonstrate that SQuBa successfully combines LLM summarization quality with end-to-end processing.

Table 3: **Comparison of average time per sample for generating summaries from speech. Cascaded** represents the cascaded system combining Whisper-Large-v2 and Mamba-2.8B-Zephyr.

| Model | LLM | Avg. Time per Sample (seconds) |
|---|---|---|
| Cascaded | Mamba-2.8b | 41.7 |
| Kang & Roy (2024) | MiniChat2-3b | 13.0 |
| SQuBa | Mamba-2.8b | **2.4** |

Table 4: **WER(%) comparison of different query configurations of SQuBa on LibriSpeech test sets.** Query Len. denotes the number of learnable queries representing 0.33 seconds of speech. Libri-clean and Libri-other denote WER(%) for each test subset, while Libri-all represents the average WER(%) across both subsets. Whisper Large v2 serves as our baseline speech encoder.

| Model | Query Len. | Libri-clean | Libri-other | Libri-all |
|---|---|---|---|---|
| Whisper | – | 2.87 | 5.16 | 4.42 |
| SQuBa | 1 | 5.89 | 7.81 | 6.85 |
| | 2 | **4.55** | **6.50** | **5.52** |
| | 4 | 4.80 | 7.74 | 6.80 |
| | 6 | 4.97 | 7.85 | 6.92 |
| | 8 | 5.09 | 7.94 | 7.02 |

**Speed Results** Another key aspect of our approach is the inference speed of the Mamba model, which powers our SQuBa model. To evaluate this, we conducted experiments on 400 data samples, measuring the average time required by each model to generate summaries from speech inputs.

As shown in Table 3, our proposed SQuBa model significantly outperforms transformer-based counterparts in terms of inference speed. Notably, SQuBa achieves a $17\times$ speedup compared to the cascaded pipeline, a widely adopted approach in speech summarization. This significant speedup stems from replacing the Transformer-based Whisper decoder's quadratic complexity for intermediate transcription with Mamba LLM's linear complexity throughout the pipeline, making it particularly efficient for long-form speech processing up to 6 minutes in length.

While many researchers continue to pursue cascaded summarization summarization, we highlight that adopting an end-to-end approach with the Mamba model architecture results in substantially faster inference speeds, while still delivering comparable results. This demonstrates the advantage of our approach, not only in processing time but also in scalability for real-world applications, making SQuBa a more practical solution for reliable and fast speech summarization.

## 7.2 ABLATION STUDIES

We conduct ablations on learnable query length and bootstrapped DPO. See Appendix D.1 for more ablation results on the unidirectional Mamba layer, downsampling approaches, and long speech alignment training.

### 7.2.1 QUERY SIZE

The query length in the Mamba-based projector plays a significant role in determining how efficiently the model can extract relevant features from the speech input. Several query lengths were experimented with on the Librispeech dataset, and evaluated the model's performance on the test-clean and test-other datasets.

As shown in Table 4, a query length of **2** was found to achieve the optimal balance between preserving contextual information and maintaining processing efficiency. This length allows the model to retain the necessary contextual relationships in speech while ensuring that the feature representation remains compact enough for efficient computation. When the query length is increased, the model tends to generate disconnected fragments, disrupting the natural flow of the speech signal and leading to a loss of coherence. Conversely, reducing the query length too much results in oversimpli-

Table 5: **Ablation studies on bootstrapped DPO training.** 'X' denotes SQuBa without DPO training. Both models were trained using our two-stage training framework.

| Model | DPO | ROUGE-1 | ROUGE-2 | ROUGE-L | METEOR |
|-------|-----|---------|---------|---------|--------|
| SQuBa | x | 27.5 | 10.1 | 19.8 | 30.1 |
|  | ✓ | **31.5** | **10.6** | **20.6** | **33.2** |

fication of the acoustic signal, as too much information is packed into a single query, which hampers the model's ability to interpret and process the speech effectively. Therefore, a query length of 2 provides the most favorable trade-off, ensuring both clarity and contextual integrity in the model's speech summarization process.

Note that query length optimization is performed during ASR alignment stage since this is when the projector learns fundamental speech-to-text mapping. While summarization may benefit from different representation characteristics than transcription, the initial alignment phase is crucial for establishing basic speech-to-text mapping capabilities. Given the sequential and computationally intensive nature of both training stages, optimizing query length during initial alignment ensures efficient development of a well-configured projector for subsequent fine-tuning.

### 7.2.2 EFFECTS OF DPO

As part of our ablation study, we evaluated the performance of our SQuBa model without DPO training. As shown in Table 5, the inclusion of DPO training led to an improvement in performance across all evaluation metrics, with a particularly notable increase in ROUGE scores. The impact of DPO on summary quality was further highlighted in the qualitative results, as detailed in Appendix D. With DPO training, the model produced more focused and coherent summaries, capturing the key information more effectively compared to the variant without DPO.

## 8 CONCLUSION

In this paper, we introduced SQuBa, an end-to-end Mamba-based speech summarization model designed to efficiently handle long speech inputs. By leveraging the Mamba architecture and querying-attention projector, SQuBa reduces the computational complexity typically seen in Transformer-based models. Our experiments show that SQuBa delivers competitive summarization performance with significant improvements in processing speed. The two-step training scheme effectively aligns speech and text, while Direct Preference Optimization (DPO) enhances the generation of concise, coherent summaries aligned with human preferences. SQuBa's architecture holds promise for applications like podcast summarization and meeting transcription, highlighting the potential of Mamba-based querying-attention in multimodal processing.

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

# A  TRAINING DETAILS

## A.1  MODELS

We use the pre-trained Whisper (Radford et al., 2023) as our speech encoder, specifically Whisper Large v2, which has 1.55 billion parameters and is trained on 680,000 hours of multilingual audio. The encoder has a 30-second input limit and an output dimension of 5120.

For the LLM backbone, we utilize the pre-trained Mamba LLM (Gu & Dao, 2023) with 2.8 billion parameters. Mamba was pre-trained on the SlimPajama dataset (Soboleva et al., 2023) (600 billion tokens), followed by instruction-tuning on UltraChat 200K (Ding et al., 2023), and fine-tuning on UltraFeedback (Cui et al., 2023) using Direct Preference Optimization (Rafailov et al., 2023).

## A.2  METRICS

**ROUGE Metrics**   ROUGE (Lin, 2004) is a set of metrics designed to evaluate automatic text summarization and machine translation by measuring the similarity between generated summaries and reference summaries. The core idea is to compute the overlap of $n$-grams (contiguous sequences of $n$ words) between the generated text and the reference text. We incorporate three ROUGE variants in our evaluation:

- **ROUGE-1**: Calculates the overlap of unigrams (individual words) between the generated and reference summaries, assessing the presence of key words.
- **ROUGE-2**: Calculates the overlap of bigrams (pairs of consecutive words), evaluating the preservation of word sequences and contextual information.
- **ROUGE-L**: Based on the Longest Common Subsequence (LCS) between the generated and reference summaries, this metric evaluates sentence-level structure and coherence by considering the order of words.

**METEOR**   METEOR (Banerjee & Lavie, 2005) is an evaluation metric originally developed for machine translation but has been effectively applied to summarization tasks due to its ability to capture semantic similarities beyond exact word matching. It evaluates the quality of the generated summary by aligning it to the reference summary and considering several factors:

- **Word Alignment**: Aligns words between the generated and reference summaries, accounting for exact matches, stem matches, and synonyms.
- **Precision and Recall**: Calculates a weighted harmonic mean of unigram precision and recall, with higher weight on recall to emphasize coverage of relevant content from the reference.
- **Fragmentation Penalty**: Applies a penalty for fragmented or disordered alignments, encouraging fluent and coherent summaries with proper word order.

# B  RELATED WORKS

## B.1  SPEECH LARGE LANGUAGE MODELS

With the rise of powerful large language models (LLMs) (Ouyang et al., 2022; Touvron et al., 2023) and their application to end-to-end vision-language models (Liu et al., 2023; OpenAI, 2024), efforts have been made to extend LLMs to speech processing. Unlike static images, audio data is length-varying and temporal, posing unique challenges for alignment with text. While some models are task-specific, such as for speech recognition (Fathullah et al., 2024), translation (Wu et al., 2023), or synthesis (Hao et al., 2023), others aim to harness the full potential of general-purpose LLMs by using multi-task instruction tuning for broader applications (Hu et al., 2024; Rubenstein et al., 2023; Das et al., 2024).

Recently, SALMONN (Tang et al., 2024) advanced the integration of diverse auditory inputs, including speech, audio events, and music. Using a dual-encoder architecture–Whisper (Radford et al., 2023) for speech and BEATs (Chen et al., 2022) for non-speech audio–SALMONN efficiently

Table 6: ChaTTS Model Configuration and Parameters.

| Component | Configuration/Parameters |
|---|---|
| Inference Temperature | 0.3 |
| Top-P Sampling | 0.5 |
| Top-K Sampling | 15 |
| Audio Sample Rate | 24000 Hz |
| Maximum Token Length | 2048 tokens |
| Laugh Parameter | 0 |
| Oral Parameter | 2 |
| Break Parameter | 6 |

handles complex auditory processing. Its window-level Q-Former enables efficient processing of variable-length audio sequences, proving effective for tasks like speech recognition, audio captioning, and joint reasoning across speech and audio inputs.

### B.2 LLM ALIGNMENT AND DIRECT PREFERENCE OPTIMIZATION

Human alignment (Ouyang et al., 2022; Bai et al., 2022) in LLMs ensures that models generate helpful and harmless responses aligned with human preferences. A common approach is Reinforcement Learning with Human Feedback (RLHF), which trains LLMs using preference functions estimated via a neural reward model. RLHF (Ouyang et al., 2022) typically involves three stages: (1) supervised fine-tuning, (2) reward model training, and (3) reinforcement fine-tuning, usually with Proximal Policy Optimization (PPO) (Schulman et al., 2017). PPO requires a reference model to limit policy divergence and a value estimation model. The multi-stage training of several models creates a bottleneck, and RLHF training can be unstable and sensitive to hyperparameters.

Direct Preference Optimization (DPO) (Rafailov et al., 2023) addresses these challenges by deriving an implicit reward function from the KL-constrained reward maximization objective, enabling direct optimization without the need for a reward model. By removing the unstable and resource-heavy reinforcement learning process, DPO offers a more stable and efficient approach to human alignment fine-tuning. However, further improvements are needed for DPO to reach the state-of-the-art performance of RLHF, leading to various adaptations and advancements in Preference Optimization (Azar et al., 2023; Ethayarajh et al., 2024; Meng et al., 2024).

## C ADDITIONAL DETAILS

### C.1 CASCADED MODEL DETAILS

For the cascaded model, to extract the intermediate transcriptions, we utilize the Whisper (Radford et al., 2023) built-in chunking algorithm through the Transformers pipeline, which efficiently handles audio inputs of arbitrary length. While Whisper has a 30-second input limitation per forward pass, the pipeline automatically manages longer inputs by setting chunk length to 30, following the standard approach recommended in the official implementation.[5]

We maintain Whisper Large v2 in its frozen pre-trained state. The Mamba-2.8B backbone in our cascaded baseline is fine-tuned on the MediaSum text dataset (as shown in Table 1 as "Mamba"). We made a deliberate choice not to apply DPO fine-tuning to the cascaded baseline, as the issue DPO addresses – the tendency toward transcription-like lengthy generations – is specific to end-to-end speech summarization models and does not manifest in pure text-to-text summarization scenarios.

---

[5] https://huggingface.co/openai/whisper-large-v2

Table 7: **Ablation studies on the effect of unidirectional Mamba in Q-Mamba projector with LibriSpeech and GigaSpeech.** Default denotes our SQuBa configuration described in the training details, while no-mamba denotes SQuBa without a unidirectional Mamba layer inside the Q-Mamba projector. Giga-60, Giga-90, and Giga-120 each denote the WER(%) for GigaSpeech test samples of 60, 90, and 120 seconds, respectively.

| Model | config | libri-clean | libri-other | giga-60 | giga-90 | giga-120 |
|-------|--------|-------------|-------------|---------|---------|----------|
| SQuBa | default | **4.5** | **6.5** | **15.3** | **16.9** | **17.8** |
| SQuBa | no-mamba | 4.8 | 9.8 | 16.5 | 22.9 | 23.9 |

## C.2 CHATTTS DETAILS

We utilize ChatTTS [6] with the configuration shown in table 6.

We conducted a human evaluation of generated speech samples using Mean Opinion Score (MOS). Five human annotators assessed 50 randomly selected speech samples, scoring them on a scale from 1 to 5, where 1 indicates completely unnatural and incoherent speech relative to the transcription, and 5 indicates completely natural and coherent speech. Our samples achieved an MOS of **4.16**, demonstrating that the generated speech closely resembles real-world speech.

# D RESULTS

## D.1 ADDITIONAL ABLATIONS

### D.1.1 EFFECT OF UNIDIRECTIONAL MAMBA

The unidirectional Mamba layer is incorporated within the learnable queries before cross-attention to establish a causal structure aligned with LLM input expectations. This design choice draws inspiration from the self-attention layer in Q-former and the causal attention layer in SEED-LLaMA, facilitating better conformity of generated speech tokens to LLM input requirements. To evaluate the impact of this architectural decision, ablation studies were conducted during the ASR alignment stage, comparing model performance with and without the unidirectional Mamba layer.

As shown in Table 7, including the unidirectional Mamba layer improves WER for LibriSpeech from 4.8% to 4.5%. This improvement is more pronounced in the GigaSpeech test set across different temporal spans (60s, 90s, and 120s), where the model consistently achieves lower WER, particularly for longer sequences. At 120 seconds, the model with Mamba maintains a WER of 17.8% compared to 23.9% without it, suggesting that the causal structure helps maintain coherence over extended temporal contexts.

### D.1.2 DOWNSAMPLING METHOD

Q-Mamba compresses speech features into semantic vectors using cross-attention and learnable queries, aiming to capture complex relationships more effectively than simple downsampling methods. To validate this design, we replaced only the cross-attention module with either average pooling or convolutional layers while maintaining other components, and evaluated performance during ASR alignment on LibriSpeech and GigaSpeech test sets.

As shown in Table 8, Q-Mamba's cross-attention mechanism with learnable queries outperforms simpler compression methods in preserving semantic information, achieving 4.5% WER on LibriSpeech compared to 4.8% and 4.7% for pooling and CNN projectors, respectively. This advantage is particularly evident in longer sequences, where Q-Mamba maintains a WER of 17.8% at

---

[6] https://github.com/2noise/ChatTTS

Table 8: **Ablation studies on downsampling method with LibriSpeech and GigaSpeech.** Default denotes our SQuBa configuration described in the training details, while AvgPool denotes downsampling with average pooling, and CNN denotes downsampling with a convolutional layer. Giga-60, Giga-90, and Giga-120 each denote the WER(%) for GigaSpeech test samples of 60, 90, and 120 seconds, respectively.

| Model | config | libri-clean | libri-other | giga-60 | giga-90 | giga-120 |
|-------|--------|-------------|-------------|---------|---------|----------|
| SQuBa | default | **4.5** | **6.5** | 15.3 | **16.9** | **17.8** |
| SQuBa | AvgPool | 4.8 | 7.9 | 17.4 | 24.5 | 25.2 |
| SQuBa | CNN | 4.7 | 11.6 | **15.2** | 20.8 | 21.0 |

Table 9: **Ablation studies on long speech alignment.** Default denotes our SQuBa configuration described in the training details, while libri-only denotes SQuBa without training on the GigaSpeech dataset. LC and LO each denote WER(%) for each test-clean and test-other subset from LibriSpeech. G60, G90, and G120 each denote the WER(%) for GigaSpeech test samples of 60, 90, and 120 seconds, respectively. For summarization, both models were fine-tuned on a synthesized summarization dataset. R1, R2, RL, and M each denote ROUGE-1, ROUGE-2, ROUGE-L, and METEOR scores, respectively. Bootstrapped DPO fine-tuning was not applied for both.

| Model | config | LC | LN | G60 | G90 | G120 | R1 | R2 | RL | M |
|-------|--------|-----|-----|------|------|------|------|------|------|------|
| SQuBa | default | **4.5** | **6.5** | **15.3** | **16.9** | **17.8** | **27.5** | **10.1** | **19.8** | **30.1** |
| SQuBa | no-libri | 4.8 | 9.8 | 16.5 | 22.9 | 23.9 | 18.8 | 3.5 | 12.3 | 14.4 |

120 seconds compared to 25.2% and 21.0% for average pooling and CNN projectors, respectively, demonstrating better preservation of semantic information over extended temporal contexts.

### D.1.3 EFFECT OF LONG SPEECH ALIGNMENT

Since the speech summarization involves long-form speech, it is expected that the model only trained on LibriSpeech (short audio) should perform poorly on long speech summarization. To validate this hypothesis, we conducted ablation studies comparing our full model without DPO training against a variant trained without the GigaSpeech dataset.

As shown in Table 9, while both configurations achieve comparable performance on LibriSpeech test-clean (4.5% vs 4.8% WER), the performance gap widens significantly for longer inputs. On GigaSpeech test samples, the WER of the libri-only model deteriorates substantially as input length increases, reaching 23.9% at 120 seconds compared to 17.8% for the full model. This degradation in transcription quality directly impacts summarization performance, with the libri-only model showing markedly lower scores across all metrics (e.g., ROUGE-1 drops from 27.5 to 18.8, METEOR from 30.1 to 14.4). These results demonstrate that incorporating long-form speech during alignment training is crucial for developing models capable of effective speech summarization.

### D.2 QUALITATIVE RESULTS

Qualitative results from SQuBa with and without DPO can be seen in Table 10 and Table 11.

Table 10: Summary Results from SquBa

| Model | Text |
|---|---|
| Input Transcript | The World Cup file that finally is just around the corner, just a little over two days to wait. On Sunday France and Croatia face off for the championship. Until then, however, we have enough memes, those viral images to keep us occupied. ẄORLD SPORTSD̈on Riddell breaks down the best one so far. The World Cup has captured football fans all over the world bringing moments of delirium and plenty of heartbreak. And in turn, the fans have captured the World Cup, its highs, and its lows online in memes. As the most expensive player in the world, Neymar was always going to be under the microscope but the Brazilian star brought joy to millions in a way that he could never have imagined. Clips of his overly theatrical rolling around went viral. In training, young kids all learned to do with the Neymar. Neymar! And in general video editors were having a ball even turning him into one. It's not just the players who produce the magic, the fans do too. This clip has been viewed more than 20 million times. A grandma blessing the Mexico players before their win against Germany, granddaughter Paula tweeted I'm 100 percent convinced my grandma was the reason Mexico won. Later on, Mexican supporters were sure that South Korea was the reason that they were still in the tournament. After Korea's unexpected win against Germany, random Koreans were hoisted onto shoulders and paraded through the streets like gods. South Korea's Consul General in Mexico found himself to be the guest at a most unexpected celebration England surprise run to the semi-finals inspired all kinds of creative musing... |
| Ground Truth Summary | France and Croatia face off in the World Cup final on Sunday. Fans have created memes about the tournament, featuring Neymar's falls, a grandma's blessing of Mexico players, England fans' excitement, and a satirical video by Croatian firefighters who missed a quarter-final shootout. |
| SQuBa w/o DPO | The World Cup coverage continues, with football fans all over the world captivated by the tournament. The Brazilian star, Neymar, brought moments of delight to millions of fans, with his over-the-top antics. In training, young kids all learned to do the same. The most expensive player in the world, Neymar, was always going to be under the microscope. But Brazilian fans were also responsible for the hype, with fans turning the tables and turning the fans into the stars. This clip has been viewed more than 20 million times. A grandma blessing the Mexico-Germany match even inspired her granddaughter to do the same with her own match. Fans have even turned the World Cup coverage into a meme, with a whole Neymar-inspired meme. And the World Cup coverage continues, with more coverage of the semi-finals and the final. |
| SQuBa | The World Cup file is still in the news, with daily World Cup memes and viral images capturing moments of excitement and joy. The Brazilian star Neymar was always going to be under the microscope, but fans have brought joy to millions in the form of memes and over-the-top performances. The Brazilian star's nephew, Neymer, even turned the tables on fans by tweeting a photo of himself as Neymar. The World Cup coverage continues, with fans eagerly awaiting the tournament's final. |

Table 11: Summary Results from SquBa

| Model | Text |
|-------|------|
| Input Transcript | NPR's business news starts with a new and bigger bailout for AIG. The Treasury and the Federal Reserve today announced a new bailout package for the troubled insurance giant AIG. The Federal Reserve has already extended the company more than $120 billion in funding. That hasn't stabilized it. Now the Fed and the Treasury have come up with a new, rejiggered rescue package that replaces previous measures. It includes lower interest rates on loans and the purchase of $40 billion in AIG shares. The total cost of this new AIG bailout is about $150 billion. |
| Ground Truth Summary | The Treasury and Federal Reserve announced a new $150 billion bailout package for AIG, replacing previous measures with lower interest rates on loans and the purchase of $40 billion in AIG shares. The total cost is about $150 billion. |
| SQuBa w/o DPO | The Treasury and the Federal Reserve today announced a new and bigger bailout package for the troubled insurance giant, AIG. The Federal Reserve has already extended the company by more than $1.2 trillion in funding, but it hasn't stabilized it. So now the Federal Reserve and the Treasury have come up with a new rescue package that includes lower interest rates on loans and the purchase of $40 billion in AIG shares. The total cost of this new bailout package is about $170 billion. |
| SQuBa | The Treasury and Federal Reserve have announced a new bailout package for the troubled insurance giant, AIG. The package includes lower interest rates on loans and the purchase of $40 billion in AIG shares. The cost of the new bailout package is not specified, but it has been extended by more than $1.2 trillion in funding already. |

