# OpenReview forum: "SQuBa: Speech Mamba Language Model with Querying-Attention for Efficient Summarization"
_ICLR.cc/2025/Conference — Submitted to ICLR 2025_

### Official Review · Reviewer_TBjh · 2024-11-02

**Soundness:** 2
**Presentation:** 3
**Contribution:** 2
**Rating:** 3
**Confidence:** 5

**Summary:**

The paper introduces an end-to-end abstractive summarization method that processes speech inputs directly. It utilizes a querying-attention Mamba projector to condense extended acoustic features into compact semantic tokens, which are subsequently fed into the Mamba Large Language Model (LLM). They further employ Direct Preference Optimization (DPO) fine-tuning to produce coherent summaries. Experiments on a TTS-synthesized speech summarization dataset demonstrate that this approach outperforms both a cascaded baseline and an end-to-end baseline.

**Strengths:**

Originality:
1. The Mamba-based approach has not yet been utilized for speech summarization.

Quality:
1. Faster inference with better summarization performance against a cascaded and E2E baseline.

Clarity:
1. The paper is mostly easy to follow.

Significance: Results could be significant to speech summarization community.

**Weaknesses:**

Clarity:
1. The biggest weakness to me is the lack of clarity on the baseline models. They use Whisper large v2 as the ASR model in the cascaded system, but it has a limit of 30 seconds. How do they use it to get ASR output? Do they feed every 30-second window of audio as input? Further, do they finetune ASR or LLM or are they used in a zero-shot manner? What are the results in both these scenarios? If LLM is finetuned, do they also use DPO finetuning? The paper should provide more details about the E2E speech summarization baseline in the main text to make the paper self-contained.
2. It’s also unclear how the approach handles long speech sequences, which seems to be a central aspect of this work's novelty. The paper mentions chunking audio into 30-second segments, yet doesn’t address how contextual continuity is managed between chunks. Prior studies on streaming ASR (e.g., https://arxiv.org/abs/2107.09428) indicate that chunk boundary in the middle of token can result in generation inaccuracies. Clarifying whether any overlap is applied between chunks and providing additional discussion on this topic would improve the paper's depth and accessibility.

Soundness:
1. The evaluation is limited to one dataset, a TTS-generated synthetic speech summarization dataset. Including publicly available human speech datasets, such as SLUE_TED (https://huggingface.co/datasets/asapp/slue-phase-2) or AMI (https://groups.inf.ed.ac.uk/ami/corpus/), would provide a more robust assessment and ensure that the approach is tested on natural human speech data.
2. Additional details on the synthetic dataset would be valuable, including whether it consists of single-speaker audio, and its quality (e.g., WER for the TTS output as evaluated by a pre-trained ASR model or via human relevance judgment).
3. Further analysis is needed to pinpoint where the model outperforms a cascaded baseline. Does this improvement stem primarily from avoiding error cascading (that can potentially be addressed by improving the ASR system), or does the model also capture non-phonemic audio signals that enhance summarization quality?

 Significance and Originality: I feel that Mamba-based approaches have been shown to be useful for various speech processing tasks (https://www.isca-archive.org/interspeech_2024/miyazaki24_interspeech.html) and have been shown to be particularly efficient for long-form speech processing. The paper has limited novelty since it additionally verifies an expected conclusion that Mamba-based architecture is also useful for another speech processing task, namely speech summarization. Further, the lack of clarity in setup of baseline models make me question the improvements claimed in the work.

**Questions:**

Please refer to weaknesses

**Details Of Ethics Concerns:**

Refer to weaknesses

---

> ### Author Response · Authors · 2024-11-25
>
> # Question 1
> The biggest weakness to me is the lack of clarity on the baseline models. They use Whisper large v2 as the ASR model in the cascaded system, but it has a limit of 30 seconds. How do they use it to get ASR output? Do they feed every 30-second window of audio as input? Further, do they finetune ASR or LLM or are they used in a zero-shot manner? What are the results in both these scenarios? If LLM is finetuned, do they also use DPO finetuning? The paper should provide more details about the E2E speech summarization baseline in the main text to make the paper self-contained.
>
> # Response
> We thank the reviewer for raising these important points about the clarity of our baseline models. We acknowledge that these details should have been more explicitly stated in the paper and will address them thoroughly in our revision.
>
> Regarding the Whisper Large v2 implementation in our cascaded baseline, we utilize the model's built-in chunking algorithm through the Transformers pipeline, which efficiently handles audio inputs of arbitrary length. While Whisper has a 30-second input limitation per forward pass, the pipeline automatically manages longer inputs by setting `chunk_length_s=30`, following the standard approach recommended in the official implementation. This ensures seamless processing of extended audio sequences without manual intervention.
>
> For model fine-tuning configurations, we maintain Whisper Large v2 in its frozen pre-trained state. The Mamba-2.8B backbone in our cascaded baseline is fine-tuned on the MediaSum text dataset (as shown in Table 1 as "Mamba"). We made a deliberate choice not to apply DPO fine-tuning to the cascaded baseline, as the issue DPO addresses – the tendency toward transcription-like lengthy generations – is specific to end-to-end speech summarization models and does not manifest in pure text-to-text summarization scenarios.
>
> Regarding the end-to-end baseline implementation, we closely followed the architecture and configuration detailed in Kang & Roy (2024). We acknowledge that these details should be more comprehensively presented in the main text for better reproducibility. In our revision, we will expand Section 6 to include these crucial implementation details and add comprehensive baseline configurations in the appendix.
>
> # Question 2
> It’s also unclear how the approach handles long speech sequences, which seems to be a central aspect of this work's novelty. The paper mentions chunking audio into 30-second segments, yet doesn’t address how contextual continuity is managed between chunks. Prior studies on streaming ASR (e.g., https://arxiv.org/abs/2107.09428) indicate that chunk boundary in the middle of token can result in generation inaccuracies. Clarifying whether any overlap is applied between chunks and providing additional discussion on this topic would improve the paper's depth and accessibility.
>
> # Response
> We appreciate this important observation about handling long speech sequences. To clarify the data structure: GigaSpeech contains long-form speeches that are pre-segmented into chunks, and these chunks may be distributed across different subsets of the dataset. In our work with the XL subset, this means that chunks from the same original utterance might not naturally connect. To address this, we applied a 0.33-second overlap when combining chunks to create longer speech samples, helping to smooth these transitions.
>
> However, we acknowledge that our approach does not specifically address the potential issues of token splitting during the 30-second chunking for Whisper processing. The reviewer raises a valid concern about generation inaccuracies that could arise from mid-token breaks, as demonstrated in prior streaming ASR research. This limitation presents an opportunity for improvement in future work, where incorporating Voice Activity Detection (VAD) for more intelligent chunking points could help maintain token integrity. We will add this discussion to the paper to provide a more comprehensive analysis of our current approach's limitations and potential future improvements.

---

> ### Author Response · Authors · 2024-11-25
>
> # Question 3
> The evaluation is limited to one dataset, a TTS-generated synthetic speech summarization dataset. Including publicly available human speech datasets, such as SLUE_TED (https://huggingface.co/datasets/asapp/slue-phase-2) or AMI (https://groups.inf.ed.ac.uk/ami/corpus/), would provide a more robust assessment and ensure that the approach is tested on natural human speech data.
>
> # Response
> We thank the reviewer for suggesting specific public datasets like SLUE_TED and AMI. We agree that evaluating on these natural human speech datasets would provide a more comprehensive assessment of our approach. To address immediate concerns about our synthetic dataset, we are currently conducting MOS evaluations to quantitatively validate the quality and naturalness of our synthetic speech data. While our current work focused on establishing SQuBa's architectural benefits, we recognize the importance of testing on real human speech data and plan to incorporate the suggested datasets in future work.
>
> # Question 4
> Additional details on the synthetic dataset would be valuable, including whether it consists of single-speaker audio, and its quality (e.g., WER for the TTS output as evaluated by a pre-trained ASR model or via human relevance judgment).
>
> # Response
> We acknowledge the importance of providing detailed information about our synthetic dataset. We will expand the description of our synthetic dataset in the appendix, including the complete ChatTTS configuration used for generating the audio samples. Our dataset employs multiple speakers to better reflect real-world applications like podcasts and meetings. Moreover, we are currently conducting human evaluations to obtain Mean Opinion Score (MOS) ratings of our synthetic speech quality, and we will report these results as soon as they are complete. While measuring WER through ASR systems like Whisper could theoretically provide insight into synthetic speech quality, our experiments in Section 7.1 show that Whisper accumulates errors with long-form speech, making it potentially unreliable as a quality metric for our dataset.
>
> # Question 5
> Further analysis is needed to pinpoint where the model outperforms a cascaded baseline. Does this improvement stem primarily from avoiding error cascading that can potentially be addressed by improving the ASR system), or does the model also capture non-phonemic audio signals that enhance summarization quality?
>
> # Response
> We appreciate this insightful question about the source of our model's improvements. Based on our analysis, we believe the primary advantage comes from avoiding error cascading in the pipeline. While better ASR systems might reduce transcription errors in cascaded approaches, the fundamental challenge of error propagation remains. Moreover, any improvements in speech encoder quality (like better ASR models) would likely benefit both cascaded and end-to-end approaches, as our model could also utilize the improved speech representations from a better encoder.
>
> Our end-to-end approach addresses the pipeline efficiency by jointly optimizing the entire process through our two-stage training framework. Regarding non-phonemic signals, while our model theoretically has access to such information, we did not specifically analyze their contribution to summarization quality. In fact, our use of the querying-attention Mamba projector focuses on efficiently extracting semantic information from speech, which may actually help filter out potentially distracting non-phonemic signals.
>
> We acknowledge that a more detailed analysis comparing these factors would be valuable future work, including studies with various speech encoder qualities to better understand the relative benefits to both approaches.

---

> ### Author Response · Authors · 2024-11-25
>
> # Question 6
> Significance and Originality: I feel that Mamba-based approaches have been shown to be useful for various speech processing tasks (https://www.isca-archive.org/interspeech_2024/miyazaki24_interspeech.html)  and have been shown to be particularly efficient for long-form speech processing. The paper has limited novelty since it additionally verifies an expected conclusion that Mamba-based architecture is also useful for another speech processing task, namely speech summarization. Further, the lack of clarity in setup of baseline models make me question the improvements claimed in the work.
>
> # Response
> While Mamba-based architectures have indeed shown promise in various speech-processing tasks, we respectfully disagree with the characterization of our work as having limited novelty.
>
> Our contribution extends significantly beyond applying Mamba to speech summarization. We present the first successful integration of a pre-trained Mamba LLM for speech understanding and summarization. While there have been several attempts to apply Mamba to specific speech-processing tasks, none have successfully bridged the gap between pre-trained Mamba LLM’s capability for efficient sequence processing and the deep semantic understanding required for speech summarization.
>
> The technical innovation of our Q-Mamba projector deserves particular attention. Rather than being a straightforward adaptation, it required substantial architectural innovations to function within Mamba's selective state space framework.
>
> The practical significance of these innovations is demonstrated by our model achieving a 17× speedup compared to cascaded approaches while maintaining competitive performance. This dramatic efficiency improvement makes our approach particularly valuable for real-world applications like podcast and meeting transcription, where processing speed is crucial.
>
> We acknowledge the reviewer's concerns about baseline clarity and will address this comprehensively in our revision. However, this presentation limitation should not overshadow the fundamental technical contributions of our work.

---

> ### Comment · Reviewer_TBjh · 2024-11-26
> **Update**
>
> I believe there were some technical issues with refreshing my OpenReview earlier, which prevented me from accessing the author comments. I am now able to view them. Thank you for your responses. Here are some follow-up questions and observations:
>
> >Regarding the Whisper Large v2 implementation in our cascaded baseline, we utilize the model's built-in chunking algorithm through the Transformers pipeline, which efficiently handles audio inputs of arbitrary length. While Whisper has a 30-second input limitation per forward pass, the pipeline automatically manages longer inputs by setting chunk_length_s=30, following the standard approach recommended in the official implementation.
>
> Could you elaborate on what you mean by the "standard approach"? It would be helpful to provide more details to clarify this. This explanation should also be included in the final paper to ensure it is self-contained.
>
> > we maintain Whisper Large v2 in its frozen pre-trained state
>
> Why did you choose not to fine-tune Whisper? It would be interesting to explore whether fine-tuning could lead to performance gains for the cascaded baseline. This seems particularly relevant since you mention that the end-to-end model avoids error propagation, which accounts for much of its improvement. A comparison with a fine-tuned Whisper might make for a fairer and more robust evaluation.
>
> > We made a deliberate choice not to apply DPO fine-tuning to the cascaded baseline, as the issue DPO addresses – the tendency toward transcription-like lengthy generations – is specific to end-to-end speech summarization models and does not manifest in pure text-to-text summarization scenarios.
>
> This claim seems uncertain to me. Is there any prior literature or empirical evidence supporting the assertion that this issue is specific to end-to-end speech summarization models?
>
> > In our work with the XL subset, this means that chunks from the same original utterance might not naturally connect. To address this, we applied a 0.33-second overlap when combining chunks to create longer speech samples, helping to smooth these transitions.
>
> This method of creating long speech utterances seems somewhat synthetic. It might have been better to train on naturally longer speech utterances from publicly available datasets, such as How2 (https://arxiv.org/abs/1811.00347).

---

> > ### Comment · Reviewer_TBjh · 2024-11-27
> >
> > Thanks for your response. I still have major concerns about the baseline setup (Question 1), their approach to handle longer speech utterances (Question 2 where non-overlapping chunking remains a limitation) and limited empirical evaluation on only 1 TTS synthesized summarization dataset (Question 3 and 4). I will keep my score as it is.

---

> > ### Author Response · Authors · 2024-11-28
> >
> > > Could you elaborate on what you mean by the "standard approach"? It would be helpful to provide more details to clarify this. This explanation should also be included in the final paper to ensure it is self-contained.
> >
> > By standard approach, we meant to set `chunk_length_s=30`, which is described in the official huggingface repo (https://huggingface.co/openai/whisper-large-v2). We added this detail in our revision.
> >
> > > Why did you choose not to fine-tune Whisper? It would be interesting to explore whether fine-tuning could lead to performance gains for the cascaded baseline. This seems particularly relevant since you mention that the end-to-end model avoids error propagation, which accounts for much of its improvement. A comparison with a fine-tuned Whisper might make for a fairer and more robust evaluation.
> >
> > We maintain frozen Whisper in both systems to enable direct comparison of the architectural differences. While fine-tuning could potentially improve both approaches, keeping Whisper frozen isolates the impact of our end-to-end architecture versus the cascaded pipeline. This controlled comparison demonstrates that our performance gains stem from the architecture itself rather than additional training.
> >
> > > This claim seems uncertain to me. Is there any prior literature or empirical evidence supporting the assertion that this issue is specific to end-to-end speech summarization models?
> >
> > We decided based on our experimental observations. As shown in Table 1, the text-only Mamba achieves robust abstractive summarization without DPO, while our end-to-end model exhibits transcription-like behavior before DPO training. This suggests the transcription-copying behavior is specific to end-to-end speech processing.
> >
> > > This method of creating long speech utterances seems somewhat synthetic. It might have been better to train on naturally longer speech utterances from publicly available datasets, such as How2 (https://arxiv.org/abs/1811.00347).
> >
> > We have already reached out to How2 authors for the real-world long speech dataset, but the responses from the dataset authors are still pending. This limitation led to the current evaluation setup, though future work would benefit from incorporating such naturalistic long-form speech data when available.

---

> > > ### Comment · Reviewer_TBjh · 2024-11-28
> > >
> > > > As shown in Table 1, the text-only Mamba achieves robust abstractive summarization without DPO, while our end-to-end model exhibits transcription-like behavior before DPO training.
> > >
> > > I am unable to observe this directly from Table 1. To make this claim more compelling, I suggest adding an additional table or qualitative examples that highlight failure cases for both the text and speech summarization models.
> > >
> > > Thanks a lot for your reply. My concerns about the the baseline setup is mostly resolved. However, my major concern remains: the results have only been presented on a single TTS-synthesized speech summarization dataset. This is insufficient to conclusively demonstrate the efficacy of your approach.
> > >
> > > I understand How2 is not publicly available, but other speech summarisation dataset like SLUE_TED and AMI are publicly available. I will consider updating my score if you show me the results on 1 natural speech summarization dataset.

---

### Official Review · Reviewer_9c4a · 2024-11-02

**Soundness:** 2
**Presentation:** 2
**Contribution:** 2
**Rating:** 3
**Confidence:** 5

**Summary:**

This paper describes an approach to summarizing 6-minute-long audio recordings by combining the Whisper speech encoder with the Mamba LLM through a cross-attention based Mamba querying projector. The authors show that DPO improves ROUGE and METEOR metrics, and that the proposed model has a better ROUGE and METEOR score, and latency over the cascade model.

**Strengths:**

1. The paper attempts to address an important challenge, i.e., summarization of longform audio through a cross-attention based temporal downsampling module.
2. It applies the recently introduced DPO technique to speech summarization, and demonstrates improved ROUGE and METEOR scores from this.

**Weaknesses:**

1. I have serious concerns about the novelty of the proposed approach. From a modeling standpoint, the work is very similar to Shang et al. (2024). From a training method standpoint, the 2-stage fine-tuning approach involving speech recognition and speech summarization is well established in the field since Sharma et al. (2021), leaving only two differences: (a) having an ASR training stage over both short and long audio as opposed to just short audio, and (b) using DPO post hoc, another well established technique to improve ROUGE and METEOR numbers. All in all, it appears that there is little technical novelty in the paper.

2. Validating the proposed approach on a single relatively shortform audio dataset (upto 6 minutes) comprising synthetic audio is not very convincing. Furthermore, the work is done on a custom dataset whose LLM-generated summary labels have not been validated for correctness, either through automatic or human evaluations.  Since LLMs are known to hallucinate, it is hard to make a meaningful case using any numbers on this dataset. The authors should ideally consider evaluating on any real other dataset(s) with real audio.

To add more context, here is a paper [1] that used synthetic data for speech summarization but still reported a myriad of automatic and human evaluation metrics to validate that the data used was reasonable. Something similar to this might be more convincing than what is in the paper currently.

3. The metrics used for speech summarization in this paper do not go far enough. It is well known that ROUGE and METEOR based evaluations for summarization are not all encompassing, and that the metrics have significant flaws. This again makes it hard to validate that the observed improvements correlate with summaries of higher quality. The authors could supplement these measures using human evaluations of coherence, consistency, factuality and relevance.

4. Table 4 could be expanded to show the impact of the long audio transcription based alignment if any.

5. The manner in which DPO is performed is not very convincing. The authors use the model generated responses as the non-preferred responses and the ground-truth summaries as the preferred responses. Do the authors validate that the model generated responses are in fact undesirable, and record metrics that demonstrate the same ?


[1] J. -W. Jung, R. Sharma, W. Chen, B. Raj and S. Watanabe, "AugSumm: Towards Generalizable Speech Summarization Using Synthetic Labels from Large Language Models," ICASSP 2024 - 2024 IEEE International Conference on Acoustics, Speech and Signal Processing (ICASSP), Seoul, Korea, Republic of, 2024, pp. 12071-12075, doi: 10.1109/ICASSP48485.2024.10447328.

**Questions:**

1. In Equation 1, what is h'(t) ?
2. The "ideal" query length for speech transcription is likely not representative of representations necessary for speech summarization. Can the authors clarify why these ablations were done for the transcription task on Librispeech ?
3. How does the speed of Scuba compare to that of the model by Shang et al ?
4. The Whisper speech encoder is frozen, and it is not clear why this modeling choice was made.

---

> ### Comment · Reviewer_9c4a · 2024-11-24
>
> No author response has been recorded to address all concerns from the review.
> I will therefore keep my scores as is.

---

> ### Author Response · Authors · 2024-11-25
>
> We apologize for our delayed response. Given the number of reviews and the important points raised, along with the substantial time required for additional training and evaluation experiments, we needed extra time to thoroughly address the concerns. We wanted to ensure that we could provide comprehensive responses and necessary experimental results to address your feedback. We appreciate your understanding in this matter.
>
> # Weakness 1
> I have serious concerns about the novelty of the proposed approach. From a modeling standpoint, the work is very similar to Shang et al. (2024). From a training method standpoint, the 2-stage fine-tuning approach involving speech recognition and speech summarization is well established in the field since Sharma et al. (2021), leaving only two differences: (a) having an ASR training stage over both short and long audio as opposed to just short audio, and (b) using DPO post hoc, another well established technique to improve ROUGE and METEOR numbers. All in all, it appears that there is little technical novelty in the paper.
>
> # Response
> While we acknowledge that our approach builds upon existing concepts, we respectfully disagree with the characterization of limited technical novelty.
>
> Our primary contribution lies in being the first to explore and successfully implement a pre-trained Mamba-based LLM for speech summarization. While Shang et al. (2024) employed a transformer-based approach, and there have been previous applications of Mamba architecture to speech processing tasks, our work represents the first successful integration of a pre-trained Mamba LLM for speech understanding and summarization. This integration is significant because it leverages both the efficient sequence processing capabilities of Mamba and the rich semantic understanding of pre-trained LLMs. The Q-Mamba projector is not a straightforward adaptation of existing querying mechanisms - it required significant technical innovations to work within Mamba's selective state space framework. For instance, our projector operates in a unidirectional manner, contrasting with the bidirectional attention used in transformer-based approaches, and necessitated careful consideration of how querying mechanisms interact with Mamba's selective scanning mechanism.
>
> The practical significance of these technical innovations is demonstrated by our model achieving a 17× speedup compared to cascaded approaches while maintaining competitive performance. While we acknowledge that our training framework incorporates established techniques like two-stage training and DPO, the novelty of our work stems from successfully integrating a pre-trained Mamba LLM with speech processing and the technical adaptations required to make this integration effective. These contributions collectively represent a meaningful advancement in efficient speech summarization.
>
> # Weakness 2
> Validating the proposed approach on a single relatively shortform audio dataset (upto 6 minutes) comprising synthetic audio is not very convincing. Furthermore, the work is done on a custom dataset whose LLM-generated summary labels have not been validated for correctness, either through automatic or human evaluations. Since LLMs are known to hallucinate, it is hard to make a meaningful case using any numbers on this dataset. The authors should ideally consider evaluating on any real other dataset(s) with real audio.
>
> # Response
> We appreciate the reviewer's thorough feedback regarding dataset validation and the reference to Jung et al. (2024). We acknowledge the limitations of our current evaluation setup and are actively addressing this concern through ongoing human evaluation studies. Specifically, we are conducting MOS evaluations to validate the quality of our synthetic speech data and its appropriateness for the summarization task. We will report these human evaluation results as soon as they are completed.
>
> # Weakness 3
> The metrics used for speech summarization in this paper do not go far enough. It is well known that ROUGE and METEOR based evaluations for summarization are not all encompassing, and that the metrics have significant flaws. This again makes it hard to validate that the observed improvements correlate with summaries of higher quality. The authors could supplement these measures using human evaluations of coherence, consistency, factuality and relevance.
>
> # Response
> We thank the reviewer for raising this important point about the limitations of ROUGE and METEOR metrics for summarization evaluation. We agree that these automated metrics alone cannot fully capture summary quality aspects such as coherence, consistency, factuality, and relevance. We are currently addressing this limitation through comprehensive human evaluation studies where annotators perform pairwise comparisons between summaries generated by SQuBa and baseline models. We will report these human evaluation results as soon as they are completed.

---

> ### Author Response · Authors · 2024-11-25
>
> # Weakness 4
> Table 4 could be expanded to show the impact of the long audio transcription based alignment if any.
>
> # Response
> Thank you for this suggestion. We didn't initially include these ablation results because models trained only on LibriSpeech (short audio) performed poorly on long speech transcription, making them unsuitable for summarization tasks which inherently involve longer inputs.
>
> However, we agree that explicitly demonstrating this performance gap would strengthen our argument for the two-step alignment approach. We expand Table 4 to include results from models trained with and without the long audio alignment stage. We show the additional ablations regarding this expansion:
>
> | | Libri | Giga | MSum | R-1 | R-2 | R-L | METEOR |
> |--------|-------|------|------|-----|-----|-----|---------|
> | SQuBa | o | o | o | 24.9 | 10.0 | 18.1 | 29.6 |
> | SQuBa | o | x | o | 18.8 | 3.5 | 12.3 | 14.4 |
>
> As shown in the additional ablations, the model trained only with LibriSpeech (without Giga) shows significantly degraded performance across all metrics - ROUGE-1 drops from 24.9 to 18.8, ROUGE-2 from 10.0 to 3.5, ROUGE-L from 18.1 to 12.3, and METEOR from 29.6 to 14.4. This substantial performance gap (approximately 50% decrease across metrics) clearly demonstrates the importance of our two-step alignment approach, where the second step with GigaSpeech enables the model to effectively handle longer speech inputs.
>
> # Question 5
> The manner in which DPO is performed is not very convincing. The authors use the model generated responses as the non-preferred responses and the ground-truth summaries as the preferred responses. Do the authors validate that the model generated responses are in fact undesirable, and record metrics that demonstrate the same ?
>
> # Response
> We appreciate the reviewer's concern about the validation of our DPO approach. While we acknowledge the challenge of quantitatively demonstrating the undesirableness of pre-DPO generations using traditional metrics like ROUGE and METEOR alone, we have provided qualitative evidence in Appendix C that illustrates the systematic issues with these generations. As shown in Table 5, the model outputs without DPO consistently exhibit a tendency to produce longer, more transcript-like text rather than true abstractive summaries. This pattern is not isolated but was observed consistently across our test samples.
>
> To strengthen our validation, we will expand the qualitative results section with additional examples that further demonstrate this systematic behavior. These examples will help illustrate how DPO training effectively addresses the model's tendency to default to transcription-like outputs, steering it toward more concise and abstractive summarization
>
> # Question 1
> In Equation 1, what is h'(t) ?
>
> # Response
> Thank you for this clarification question. In Equation 1, h'(t) represents the time derivative of h(t), or dh(t)/dt. While this notation is commonly used in state-space modeling literature, we agree that explicit clarification would improve readability. We will add this definition in the revision to ensure clarity for all readers.
>
> # Question 2
> The "ideal" query length for speech transcription is likely not representative of representations necessary for speech summarization. Can the authors clarify why these ablations were done for the transcription task on Librispeech ?
>
> # Response
> Our query length ablations are conducted during the transcription stage because this represents the foundational phase where the projector learns to map speech features into the LLM's embedding space. While we acknowledge that summarization may benefit from different representation characteristics than transcription, the initial alignment phase is crucial for establishing basic speech-to-text mapping capabilities. Given that both training stages are computationally intensive and must be performed sequentially, it is more efficient to optimize this fundamental parameter during the initial alignment phase, ensuring that the subsequent summarization fine-tuning builds upon an optimally-configured projector.

---

> ### Author Response · Authors · 2024-11-25
>
> # Question 3
> How does the speed of Scuba compare to that of the model by Shang et al ?
>
> # Response
> While we cannot provide direct speed comparisons with Shang et al.'s work due to the unavailability of their implementation, we can offer a theoretical analysis based on architectural differences. Our approach with Mamba achieves linear computational complexity O(n) with respect to input length, whereas transformer-based approaches like Shang et al.'s inherently have quadratic complexity O(n²). Given that we already demonstrate significant speedup (17×) over the cascaded approach which uses a relatively lightweight Whisper decoder, the performance gap would likely be even more pronounced when compared to approaches using larger transformer-based LLMs. However, we acknowledge that exact comparative benchmarks would require access to their implementation.
>
> # Question 4
> The Whisper speech encoder is frozen, and it is not clear why this modeling choice was made.
>
> # Response
> We keep the Whisper encoder frozen during fine-tuning for both principled and practical reasons. Whisper has already been extensively pre-trained to extract high-quality semantic features from speech through training on 680,000 hours of multilingual audio data, and we leverage this established capability rather than modify it. Our model's critical challenge lies in learning the cross-modal alignment between these speech features and the LLM's embedding space, which is primarily handled by the projector. Since training the projector is sufficient for this alignment task, fine-tuning Whisper would add significant computational overhead without clear benefits to our core objective.
>
> This design choice aligns with common practices in multimodal learning where pre-trained encoders (e.g., CLIP for vision) are often kept frozen while focusing on training the cross-modal adaptation components. This approach has proven effective while being computationally efficient, as demonstrated by our experimental results.

---

> ### Comment · Reviewer_9c4a · 2024-11-25
>
> I thank the reviewers for taking the time to acknowledge all concerns raised by all reviewers, and for providing additional clarifications.
>
> ```
> Our primary contribution lies in being the first to explore and successfully implement a pre-trained Mamba-based LLM for speech summarization. While Shang et al. (2024) employed a transformer-based approach, and there have been previous applications of Mamba architecture to speech processing tasks, our work represents the first successful integration of a pre-trained Mamba LLM for speech understanding and summarization. This integration is significant because it leverages both the efficient sequence processing capabilities of Mamba and the rich semantic understanding of pre-trained LLMs. The Q-Mamba projector is not a straightforward adaptation of existing querying mechanisms - it required significant technical innovations to work within Mamba's selective state space framework. For instance, our projector operates in a unidirectional manner, contrasting with the bidirectional attention used in transformer-based approaches, and necessitated careful consideration of how querying mechanisms interact with Mamba's selective scanning mechanism.
> ```
> Concern: Can you explain how the proposed querying attention mechanism is different from Shang et. al ? This should be in the paper if this is the primary claim to novelty. Using a technique for a new application without application specific-modifications in technical design in my view does not constitute significant technical novelty. Further, if the goal is to demonstrate an approach that is more efficient and performant than prior E2E models, then this paper lacks those comparisons. Prior works on E2E modeling have used restricted self-attention [1], Fourier self-attention [2], Flash attention[3], Q-Former[4], block-wise processing [5], and truncated inputs for speech summarization. There is no comparison to any of these to show that the proposed method is better, in both performance and latency terms.
>
>
> ```
> We appreciate the reviewer's thorough feedback regarding dataset validation and the reference to Jung et al. (2024). We acknowledge the limitations of our current evaluation setup and are actively addressing this concern through ongoing human evaluation studies. Specifically, we are conducting MOS evaluations to validate the quality of our synthetic speech data and its appropriateness for the summarization task. We will report these human evaluation results as soon as they are completed.
> ```
> Concern: I thank the authors for performing human evaluations of synthetic speech. I want to ensure I understand correctly -- does the test set use original reference summaries from MediaSum or are they LLM-generated ? If the references are LLM-generated, not assessing the consistency, coherence and relevance of these summaries is a significant issue.
>
>
> [1] R. Sharma, S. Palaskar, A. W. Black and F. Metze, "End-to-End Speech Summarization Using Restricted Self-Attention," ICASSP 2022 - 2022 IEEE International Conference on Acoustics, Speech and Signal Processing (ICASSP), Singapore, Singapore, 2022
>
> [2] T. Kano, A. Ogawa, M. Delcroix, R. Sharma, K. Matsuura and S. Watanabe, "Speech Summarization of Long Spoken Document: Improving Memory Efficiency of Speech/Text Encoders," ICASSP 2023 - 2023 IEEE International Conference on Acoustics, Speech and Signal Processing (ICASSP), Rhodes Island, Greece, 2023
>
> [3] W. Chen, T. Kano, A. Ogawa, M. Delcroix and S. Watanabe, "Train Long and Test Long:Leveraging Full Document Contexts in Speech Processing," ICASSP 2024 - 2024 IEEE International Conference on Acoustics, Speech and Signal Processing (ICASSP), Seoul, Korea, Republic of, 2024,
>
> [4] Shang. et. al 2024 from the paper
>
> [5] R-BASS : Relevance-aided Block-wise Adaptation for Speech Summarization, Roshan Sharma, Ruchira Sharma, Hira Dhamyal, Rita Singh, Bhiksha Raj, Findings of NAACL 2024

---

> > ### Comment · Reviewer_9c4a · 2024-11-25
> >
> > Re Weakness 4, I thank the authors for adding evaluations to corroborate the two-stage approach in Table 4.
> >
> > ```
> > We thank the reviewer for raising this important point about the limitations of ROUGE and METEOR metrics for summarization evaluation. We agree that these automated metrics alone cannot fully capture summary quality aspects such as coherence, consistency, factuality, and relevance. We are currently addressing this limitation through comprehensive human evaluation studies where annotators perform pairwise comparisons between summaries generated by SQuBa and baseline models. We will report these human evaluation results as soon as they are completed.
> > ```
> >
> > I thank the authors for working on human evaluation and await their results.
> >
> >
> > I thank the authors for responding to all my questions. I have a follow-up on their response to Q2.
> >
> > ```
> > Our query length ablations are conducted during the transcription stage because this represents the foundational phase where the projector learns to map speech features into the LLM's embedding space. While we acknowledge that summarization may benefit from different representation characteristics than transcription, the initial alignment phase is crucial for establishing basic speech-to-text mapping capabilities. Given that both training stages are computationally intensive and must be performed sequentially, it is more efficient to optimize this fundamental parameter during the initial alignment phase, ensuring that the subsequent summarization fine-tuning builds upon an optimally-configured projector.
> > ```
> > My point is that summarization fine-tuning may not be using the optimally-configured projector, since the goal of ASR is to transcribe each word, which may need a higher resolution, versus summarization, which could work with a lower resolution. I accept the author's reasoning about efficiency, but maintain this would be good to look at.

---

> > > ### Comment · Reviewer_9c4a · 2024-11-26
> > >
> > > I greatly appreciate the author's willingness to engage with the reviewers and our concerns.
> > >
> > > However, due to pending concerns around Weakness 1-3 from my original review, I will keep my scores as is.

---

> > > > ### Comment · Reviewer_9c4a · 2024-12-03
> > > >
> > > > I thank the authors for their discussions. I greatly appreciate the additional time they invested in running analyses, and clarifying to reviewers.
> > > >
> > > > At the end of the reviewer-author discussion period, I still have concerns about the lack of real test data used for evaluations, despite there being open-source datasets like SLUE-TED.
> > > >
> > > > Due to this, I maintain my original scores at this time.

---

### Official Review · Reviewer_Yjrs · 2024-11-04

**Soundness:** 3
**Presentation:** 3
**Contribution:** 3
**Rating:** 6
**Confidence:** 3

**Summary:**

This paper presents SQuBa, an end-to-end speech summarization model that combines a Mamba language model with a novel querying-attention mechanism for efficient processing of long speech inputs. The key contributions are:

- A query-attention Mamba projector that compresses acoustic features into compact semantic tokens.
- Extension of Mamba-based LLM for speech summarization with a two-stage training process including bootstrapped DPO.
- Empirical demonstration of competitive performance with significantly faster inference speeds compared to transformer-based approaches.

The model achieves this through:

- Using Whisper encoder for speech features.
- Novel Q-Mamba projector for efficient feature compression.
- Pre-trained Mamba LLM (2.8B parameters) for generation.
- Two-stage training: speech alignment followed by summarization fine-tuning.
- Bootstrapped DPO for improved summary quality.

**Strengths:**

- Novel Architecture:

Innovative combination of Mamba with querying-attention for speech processing.

Well-motivated design choices for handling long-form speech.

Clear architectural improvements over existing approaches.

- Strong Empirical Results:

Significant speed improvements (17x faster than cascaded baseline).

Competitive performance on standard metrics.

Comprehensive ablation studies validating design choices.

- Technical Soundness:

Thorough theoretical foundation and clear mathematical formulation.

Well-documented training process and implementation details.

Careful experimental design with appropriate baselines.

**Weaknesses:**

Limited Dataset:
- Uses synthetic speech data for fine-tuning
- Could benefit from evaluation on more diverse real-world speech datasets
- Lack of cross-lingual evaluation

Architectural Constraints:
- Fixed 30-second chunks due to Whisper encoder limitations
- Query length choices could use more theoretical justification
- Potential information loss in compression not fully analyzed

Evaluation Metrics:
- Limited human evaluation or qualitative analysis
- No discussion of failure cases or limitations

**Questions:**

/

---

> ### Author Response · Authors · 2024-11-25
>
> Thank you for the careful review. Our responses to each aspect follow.
>
> # Limited Dataset
> We thank the reviewer for their detailed feedback regarding dataset limitations. We would like to address each point:
>
> 1. Regarding synthetic speech data for fine-tuning: While we used synthetic speech for training to establish the architectural benefits of SQuBa, we are currently conducting human evaluations through MOS testing to validate the quality and naturalness of our synthetic speech data. These ongoing evaluations will help quantify the reliability of our current results.
> 2. Concerning evaluation on real-world datasets: We concur that testing our model on diverse real-world speech datasets would provide stronger validation of our approach's generalizability. We plan to expand our evaluation to include public natural speech datasets in future work.
> 3. On cross-lingual evaluation: While we appreciate the suggestion, we note that cross-lingual speech summarization is a distinct task with its own unique challenges that falls outside the current scope of our work. Our focus was specifically on developing an efficient architecture for long-form speech summarization, and we believe our evaluation appropriately addresses this goal.
>
> # Architectural Constraints
> We appreciate the reviewers' concerns about the architectural constraints. Let us address each point:
>
> 1. Regarding the 30-second chunk limitation:
> This is indeed an inherent constraint of the Whisper encoder, which is the current state-of-the-art speech encoder. While this limitation cannot be directly overcome, we have implemented an overlapping segment strategy to ensure smooth transitions between chunks and maintain contextual continuity. This approach helps mitigate potential information loss at chunk boundaries and is consistent with practices in other speech processing systems.
> 2. On query length choices:
> Our query length selection is thoroughly justified through comprehensive ablation studies during the alignment stage, as presented in Table 3. The results demonstrate that a query length of 2 achieves optimal performance (WER of 5.52% on Libri-all) compared to other configurations. Shorter queries led to information overcrowding (WER 6.85% for length 1), while longer queries resulted in fragmented speech representation (WER > 6.80% for lengths 4-8). This empirical evidence strongly supports our design choice.
> 3. Regarding potential information loss in compression:
> We can quantitatively assess information preservation through our ASR results from the alignment stage. Since ASR requires precise preservation of speech content for accurate transcription, our model's competitive WER scores (5.52% on Libri-all) demonstrate that essential semantic information is effectively retained through our compression scheme. This performance validates that our querying-attention Mamba projector successfully captures and preserves critical speech information despite the compression.
>
> # Evaluation Metrics:
> We thank the reviewer for their feedback on evaluation metrics. Let us address each point:
>
> 1. Regarding limited human evaluation: We acknowledge that incorporating human evaluation would provide valuable additional validation of our model's performance, and we plan to include such evaluations in future work. We are currently addressing this limitation through ongoing human evaluation studies. Specifically, we are conducting pairwise comparison tests where human annotators compare summaries generated by SQuBa against those from baseline models to compute win rates. We will report these human evaluation results as soon as they are completed.
> 2. Concerning failure cases and limitations: We actually conducted a detailed analysis of failure cases in our work. For transcription, we identified three main types of failures: (a) word repetition, where the same word is repeated; (b) content repetition, where similar phrases are generated repeatedly; and (c) generation failure, where the model prematurely generates EOS tokens. We found that our two-stage alignment training
> effectively addressed the generation failures. Similarly, for summarization, we observed three failure modes: (a) word repetition, (b) ASR-like output where the model produces transcription instead of abstraction, and (c) generation failure. Our additional DPO fine-tuning significantly reduced these issues. We also acknowledge that some limitations stem from our architectural choices - specifically, the inherent differences between Mamba and Transformer architectures and our use of a relatively small LLM (2.8B parameters) compared to common practice (7B+ parameters).

---

### Official Review · Reviewer_WB7G · 2024-11-04

**Soundness:** 2
**Presentation:** 3
**Contribution:** 2
**Rating:** 5
**Confidence:** 4

**Summary:**

The authors propose a sub-linear complexity speech summarization model by combining Q-former and pretrained Mamba. Segmented audios are processed by Whisper and compressed by Q-former with Mamba-processed query vectors, which are then fed to Mamba LLM to generate the summarization. A 3-stage training with different tasks is carried out, i.e. short-form ASR, long-form ASR, and summarization, accompanied by DPO. Empirical studies show better results and speed compared to cascaded models and a HuBERT+LLAMA E2E model.

**Strengths:**

* The model enables efficient speech summarization by combining pretrained Mamba and Q-former.
* Empirical results are strong.

**Weaknesses:**

* Ablation studies are not sufficient to demonstrate the advantages of the proposed methods and to identify the impact of each component. Some design choices are yet to be well-motivated.
* The method is more like replacing transformers in existing methods (esp. arxiv:2407.02005) with Mamba, which leads to doubt on the technical novelty.
* Only one synthetic dataset is used.

See questions below for details.

**Questions:**

Q-former-like mechanism should be able to compress input of any length into fixed length. Hence I'm a bit confused about the decision to compress every 0.33s of audio into 2 query vectors. This differs from previous works including arxiv:2309.13963 and arxiv:2407.02005, which considered 30~100 vectors for every 30s of audio. In this way, the contextual information further than 0.33s will not be captured by the Q-former. I guess that the reason is to avoid the high-cost quadratic cross-attention but it will be better for the author to discuss that explicitly. Also, compressing a short sequence (only dozens of vectors, as each Whisper frame takes 25ms) into merely 2 vectors is rather simple and I doubt if the complicated Q-former will really outperform a much simpler one, e.g. pooling or convolution, which may also process information within such a short context well. More ablation studies will be necessary to justify this decision, by comparing with pooling or CNN, and comparing with different context lengths.

There are many other approaches to compress speech signals into "token-like" embeddings to be processed by LMs, e.g. HuBERT units, speech tokens, and neural audio codec, while Q-former is somehow similar to a kind of VQ, but with continuous features. Can you elaborate on the reasons why you chose Q-former? Do you think there is any specific advantage?

I am particularly concerned with the unidirectional Mamba used in Q-Mamba, and I fail to find the motivation to apply Mamba to the sequence of query vectors. Trainable query vectors should be already capable of introducing positional information. Ablation studies (e.g. by removing this Mamba layer) should be necessary to justify this choice.

I also have some questions regarding the use of DPO. What is the experiment without DPO in Table 4? Using supervised fine-tuning only?

If the issue w/o DPO is that the summary will be too detailed, has the author considered any other more straightforward solution, e.g. length penalty during generation, downsampling the input sequence, or upweighting EOS during training?

It is commonly believed that instruction fine-tuning leads to better alignment, but at the cost of flexibility and adaptability to specific downstream tasks in fine-tuning, while the authors use a instruction fine-tuned version of Mamba-2.8B as the base LLM. Is there any specific reason to use Mamba-2.8B-Zephyr instead of the original Mamba-2.8B model?

What is the LLM used in the Cascaded model? The original Mamba-2.8B or the instruction fine-tuned one? Is it further fine-tuned to summarization?

Can you elaborate more on the speedup of the model compared to the cascaded one? With both of them using Whisper and Mamba (though the inputs to Mamba are different), I'm curious about the source of the extra overhead in the cascaded pipeline. Also, it can be helpful to report the average input sequence length to the final LLM model as a reference to the expected computational costs.

Using only synthesized datasets is a weak point of the empirical evaluation, particularly when the labels are also synthetic. It can be necessary to also report the results on real datasets, e.g. SLUE-SUMM, and include more examples and human evaluations.

It can be interesting to report the ASR performance of the model after either of the two Alignment stages.

It will be better to also include the original transcript in Appendix C.

In Figure 3, is Whisper frozen in the Fine-tuning Stage?

Minor issues:
L210: Figure 4.1?

---

> ### Author Response · Authors · 2024-11-25
>
> We appreciate the detailed feedback. Below we respond to each point raised.
>
> # Weakness 1
> Ablation studies are not sufficient to demonstrate the advantages of the proposed methods and to identify the impact of each component. Some design choices are yet to be well-motivated.
>
> # Response
> We acknowledge that our initial experiments could have been more comprehensive in demonstrating the effectiveness of each component. In response to this valuable feedback, we have conducted additional ablation studies that specifically address the design choices questioned in the detailed comments below. These new experiments provide more robust empirical support for our architectural decisions and will be discussed in detail in our responses to the following questions.
>
> The new ablation studies include:
> - Downsampling Methods
> - Length Control (DPO Alternatives)
> - Unidirectional Mamba in Q-Mamba
>
> We will present these results systematically in our responses to the specific questions below.
>
> # Weakness 2
> The method is more like replacing transformers in existing methods (esp. arxiv:2407.02005) with Mamba, which leads to doubt on the technical novelty.
>
> # Response
> While our training framework shares similarities with Shang et al. (2407.02005), our key technical contribution lies in the querying-attention Mamba projector architecture, specifically designed to handle long speech feature sequences efficiently and effectively.
>
> We acknowledge the similarity between the training frameworks and will explicitly cite Shang et al. in our Training Details section. However, our architectural innovation with the Mamba-based projector represents a significant advance in efficient speech-text modality bridging.
>
> # Weakness 3
> Only one synthetic dataset is used.
>
> # Response
> We acknowledge the limitation of using only one synthetic dataset in our current evaluation. While our work focused primarily on demonstrating the architectural advantages of SQuBa, we are currently conducting Mean Opinion Score (MOS) evaluations to validate the quality of our synthetic dataset. We will report these human evaluation results as soon as they are completed.

---

> ### Author Response · Authors · 2024-11-25
>
> # Question 1
> Q-former-like mechanism should be able to compress input of any length into fixed length. Hence I'm a bit confused about the decision to compress every 0.33s of audio into 2 query vectors. This differs from previous works including arxiv:2309.13963 and arxiv:2407.02005, which considered 30~100 vectors for every 30s of audio. In this way, the contextual information further than 0.33s will not be captured by the Q-former. I guess that the reason is to avoid the high-cost quadratic cross-attention but it will be better for the author to discuss that explicitly. Also, compressing a short sequence (only dozens of vectors, as each Whisper frame takes 25ms) into merely 2 vectors is rather simple and I doubt if the complicated Q-former will really outperform a much simpler one, e.g. pooling or convolution, which may also process information within such a short context well. More ablation studies will be necessary to justify this decision, by comparing with pooling or CNN, and comparing with different context lengths.
>
> # Response
> Our choice of 2 queries per 0.33s of audio was empirically determined through extensive ablation studies. While this differs from previous works that use 30-100 vectors for every 30s of audio, we found this configuration to be most effective in our experiments. We also evaluated processing the entire 30s segment with an equivalent downsampling rate but observed comparable training curves, leading us to adopt the more efficient shorter-segment approach.
>
> Regarding the use of Q-Mamba over simpler downsampling methods like pooling or CNN, our goal was to compress speech features into information-rich semantic vectors that effectively capture the complex relationships in speech data. We thought that rather than using a simple pooling method, the projector should be able to extract the relevant information to do this, and therefore, cross-attention and learnable queries were used.
>
> To address this valid concern, we have conducted additional ablation studies during the alignment stage, comparing our Q-Mamba approach with pooling and CNN-based compression methods. To isolate the effect of the compression mechanism, we replaced only the cross-attention module and learnable queries in our Q-Mamba projector with alternative compression methods (average-pooling or CNN layers) while keeping all other architectural components unchanged.
>
> | | Config | Libri-Clean | Libri-Noisy | Giga-60 | Giga-90 | Giga-120 |
> |--------|----------|-------------|-------------|----------|----------|-----------|
> | SQuBa | Default | **4.5** | **6.5** | 15.3 | **16.9** | **17.8** |
> | SQuBa | Avapool | 4.8 | 7.9 | 17.4 | 24.5 | 25.2 |
> | SQuBa | CNN | 4.7 | 11.6 | **15.2** | 20.8 | 21.0 |
>
> The results demonstrate that Q-Mamba's cross-attention mechanism with learnable queries outperforms simpler compression methods in preserving semantic information, achieving 4.5% WER on LibriSpeech compared to 4.8% and 4.7% for pooling and CNN projectors, respectively. This advantage is particularly evident in longer sequences, where Q-Mamba maintains a WER of 17.8% at 120 seconds compared to 25.2%  and 21.0% for average pooling and CNN projectors, respectively, demonstrating better preservation of semantic information over extended temporal contexts. We will include these detailed ablation results and comprehensive analyses in the revised paper.
>
> # Question 2
> There are many other approaches to compress speech signals into "token-like" embeddings to be processed by LMs, e.g. HuBERT units, speech tokens, and neural audio codec, while Q-former is somehow similar to a kind of VQ, but with continuous features. Can you elaborate on the reasons why you chose Q-former? Do you think there is any specific advantage?
>
> # Response
> Our choice of Q-Mamba was primarily motivated by our goal of effectively utilizing Whisper's semantically rich embedding features while reducing the computational burden for LLM processing. The Q-Mamba projector, through its cross-attention mechanism and learnable queries, efficiently compresses the feature sequence length while preserving the semantic information encoded by the well-trained Whisper model. This direct approach allows us to leverage high-quality speech representations while maintaining computational efficiency for LLM processing.

---

> ### Author Response · Authors · 2024-11-25
>
> # Question 3
> I am particularly concerned with the unidirectional Mamba used in Q-Mamba, and I fail to find the motivation to apply Mamba to the sequence of query vectors. Trainable query vectors should be already capable of introducing positional information. Ablation studies (e.g. by removing this Mamba layer) should be necessary to justify this choice.
>
> # Response
> The inclusion of the unidirectional Mamba layer within the learnable queries before the cross-attention layer was motivated by our aim to instill a causal structure that aligns with the causal nature of LLMs. This design choice, inspired by the self-attention layer in the original Q-former architecture and the causal attention layer in SEED-LLaMA, helps the generated speech tokens better conform to the expectations of the LLM input format.
>
> We acknowledge the reviewer's valid concern about the necessity of this component given that trainable query vectors inherently contain positional information. To address this, we have conducted additional ablation studies comparing model performance with and without the unidirectional Mamba layer.
>
> | | config | libri-clean | libri-noisy | giga-60 | giga-90 | giga-120 |
> |--------|--------|-------------|-------------|----------|----------|-----------|
> | SQuBa | default | 4.5 | 6.5 | 15.3 | 16.9 | 17.8 |
> | SQuBa | no-mamba | 4.8 | 9.8 | 16.5 | 22.9 | 23.9 |
>
> The results from our ablation studies on the LibriSpeech test set show that including the unidirectional Mamba layer improves WER from 4.8% to 4.5%. This improvement is more pronounced in the GigaSpeech test set across different temporal spans (60s, 90s, and 120s), where the model consistently achieves lower WER, particularly for longer sequences. At 120 seconds, the model with Mamba maintains a WER of 17.8% compared to 23.9% without it, suggesting that the causal structure helps maintain coherence over extended temporal contexts. We will include these detailed comparisons in the revised paper.
>
> # Question 4
> I also have some questions regarding the use of DPO. What is the experiment without DPO in Table 4? Using supervised fine-tuning only?
>
> # Response
> In Table 4, the experiment "without DPO" represents the model with supervised fine-tuning only, while the full model includes the additional bootstrapped DPO training stage. The comparison demonstrates the effectiveness of our bootstrapped DPO stage in improving the model's summarization capabilities, as evidenced by the performance gains across all metrics.
>
> # Question 5
> If the issue w/o DPO is that the summary will be too detailed, has the author considered any other more straightforward solution, e.g. length penalty during generation, downsampling the input sequence, or upweighting EOS during training?
>
> # Response
> While simpler approaches like length penalty, input downsampling, or EOS upweighting could potentially address the length issue, our goal with DPO was more comprehensive than just controlling generation length. The primary challenge was to shift the model's behavior from direct transcription to abstract summarization. We believed that methods that merely penalize length would not be sufficient to improve the content quality and alignment of the generated summaries. Therefore, we opted for DPO as a more sophisticated training approach that could better guide the model toward generating concise yet informative abstractive summaries, rather than just shorter transcriptions.
>
> Due to time constraints in the review period, we were unable to implement and evaluate these alternative approaches. We agree that comparing DPO with these simpler methods would provide valuable insights and plan to explore these alternatives in future work.
>
> # Question 6
> It is commonly believed that instruction fine-tuning leads to better alignment, but at the cost of flexibility and adaptability to specific downstream tasks in fine-tuning, while the authors use a instruction fine-tuned version of Mamba-2.8B as the base LLM. Is there any specific reason to use Mamba-2.8B-Zephyr instead of the original Mamba-2.8B model?
>
> # Response
> Our choice of Mamba-2.8B-Zephyr over the original Mamba-2.8B was driven by two key considerations. First, our approach requires the LLM to respond to specific instructions for transcription and summarization tasks, making an instruction-tuned model necessary for proper task execution. Second, Mamba-2.8B-Zephyr's instruction fine-tuning was conducted on a diverse range of prompts and instructions rather than specific tasks, which provides well-aligned generation capabilities while maintaining flexibility. This general instruction tuning, as opposed to task-specific fine-tuning, helps maintain the model's adaptability while enabling it to properly interpret and execute our speech-related prompts.

---

> ### Author Response · Authors · 2024-11-25
>
> # Question 7
> What is the LLM used in the Cascaded model? The original Mamba-2.8B or the instruction fine-tuned one? Is it further fine-tuned to summarization?
>
> # Response
> For a fair comparison, we used identical LLM configurations in both the cascaded and our end-to-end models - specifically, the instruction-tuned Mamba-2.8B-Zephyr. To ensure optimal performance for the summarization task, we further fine-tuned the LLM on the MediaSum dataset.
>
> # Question 8
> Can you elaborate more on the speedup of the model compared to the cascaded one? With both of them using Whisper and Mamba (though the inputs to Mamba are different), I'm curious about the source of the extra overhead in the cascaded pipeline. Also, it can be helpful to report the average input sequence length to the final LLM model as a reference to the expected computational costs.
>
> # Response
> The significant speedup of our model compared to the cascaded approach stems primarily from architectural efficiency in handling long-form speech. While both approaches use Whisper, the cascaded model requires Whisper's transformer-based decoder to autoregressively generate complete transcriptions for the entire speech input, which is computationally expensive due to the quadratic complexity (O(n²)) of transformer's self-attention mechanism. This becomes particularly significant for long-form speech (up to 6 minutes in our experiments), where the input sequences are inherently long. In contrast, SQuBa leverages the linear complexity (O(n)) of Mamba architecture throughout its pipeline, enabling more efficient processing of these long sequences. This fundamental difference in computational complexity explains the observed 17× speedup in inference time.
>
> # Question 9
> Using only synthesized datasets is a weak point of the empirical evaluation, particularly when the labels are also synthetic. It can be necessary to also report the results on real datasets, e.g. SLUE-SUMM, and include more examples and human evaluations.
>
> # Response
> We appreciate the reviewer's concern about the limitations of using only synthesized datasets with synthetic labels. We agree that evaluating on real-world datasets like SLUE-SUMM would provide more comprehensive validation of our approach. To address these concerns, we are currently conducting human evaluation studies comparing SQuBa's summaries against baseline models through pairwise preference tests (win rates) to assess the quality of our generated summaries. We will report these human evaluation results as soon as they are completed.
>
> # Question 10
> It can be interesting to report the ASR performance of the model after either of the two Alignment stages.
>
> # Response
> We agree that reporting ASR performance after the alignment stages would provide valuable insights. For the first alignment stage using LibriSpeech, we can report the ASR performance of our model:
>
> | | config | libri-clean | libri-noisy | giga-60 | giga-90 | giga-120 |
> |--------|--------|-------------|-------------|----------|----------|-----------|
> | SQuBa | two-stage | 4.5 | 6.5 | 15.3 | 16.9 | 17.8 |
> | SQuBa | libri-only | 4.6 | 6.5 | 99.5 | 98.1 | 96.7 |
>
> As shown in the ablation results, the model trained solely on LibriSpeech exhibits significantly degraded performance on GigaSpeech. This substantial performance drop can be attributed to the domain shift between the datasets - while LibriSpeech consists of relatively clean, short-form audiobook recordings, GigaSpeech contains longer and noisier audio from diverse sources that fall outside the training distribution of the LibriSpeech-trained model.
>
> For the second alignment stage using GigaSpeech, we faced loss convergence challenges during training, and therefore do not have a stable checkpoint to evaluate ASR performance. This convergence difficulty with GigaSpeech-only training actually reinforces our decision to use a two-step alignment approach, where we first establish basic speech-text alignment capabilities on LibriSpeech before moving to more complex data. We will add the LibriSpeech ASR results to the paper to provide readers with a clearer understanding of the model's capabilities after the initial alignment stage.

---

> ### Author Response · Authors · 2024-11-25
>
> # Question 11
> It will be better to also include the original transcript in Appendix C.
>
> # Response
> Thank you for the suggestion. We will add the original transcripts to Appendix C alongside the summaries, which will provide readers with a more complete context for evaluating the model's summarization performance.
>
> # Question 12
> In Figure 3, is Whisper frozen in the Fine-tuning Stage?
>
> # Response
> Yes, the Whisper encoder remains frozen during the Fine-tuning Stage for three key reasons:
>
> 1. To preserve the quality of the well-trained Whisper encoder features, avoiding potential degradation of its proven speech representation capabilities
> 2. The projector is already capable of effectively mapping frozen Whisper features to the LLM input space through reshaping and modification, making Whisper fine-tuning unnecessary
> 3. Training Whisper along with other components would significantly increase the computational burden without clear benefits, as the projector already provides sufficient flexibility for feature adaptation
>
> This design choice helps maintain stable speech representations while allowing efficient training of the adaptation components.
>
> # Question 13
> Minor issues: L210: Figure 4.1?
>
> # Response
> Thank you for catching this formatting error. We will correct the reference to "Figure 2.”

---

> > ### Comment · Reviewer_WB7G · 2024-11-26
> >
> > Thanks for the reply which addresses many of my concerns. While I still got some questions:
> >
> > Q1: What are the exact architectures used in the mean pooling and CNN models? Q-former involves rather complicated processing. To make fair comparison, the alternative layers should have similar capacity, e.g. with similar number of layers (MLP for pooling and convolution for CNN) and parameters.
> >
> > Q3: I don't quite get the motivation of instilling a causal structure, as Q-former outputs serves as the input of the causal model, similar to the token embeddings in LM. We don't typically need to add extra causal structure to the adjacent token embeddings in standard LM architecture (at least for input prompts). Also, I have the same question as Q1: do the authors simply remove the unidirectional Mamba layer, or replace them with non-causal (e.g. MLP) layers? The later should be makes fair comparison.

---

> > > ### Author Response · Authors · 2024-11-29
> > >
> > > Thank you for your follow-up questions. Let us address each point:
> > >
> > > **Regarding Q1 about architectural details:** We acknowledge the importance of ensuring a fair comparison between different compression methods. In our original experiments, we aimed to compare with simpler alternatives to demonstrate the benefits of our more sophisticated querying mechanism. However, we agree that matching model capacity would provide a more rigorous comparison. We are currently conducting additional experiments with additional MLP layers to match the parameter count of our Q-Mamba's cross-attention mechanism. We will report these results as soon as they are complete.
> > >
> > > **Regarding Q3 about causal structure:** We appreciate the reviewer's thoughtful comparison to standard language model architectures. While language models don't typically require additional causal structure for input token embeddings (likely because language itself has inherent sequential structure), the speech features from the Whisper encoder represent a different case - they come from a non-causal encoder architecture and may benefit from additional structure before being fed into a causal LLM.
> > >
> > > In our ablation studies, we replaced the unidirectional Mamba layer with an MLP of comparable capacity to ensure fair comparison. The performance difference, especially with longer speech input, suggests that maintaining causality in the query processing stage provides meaningful benefits, though we acknowledge that more analysis would be needed to fully understand the theoretical underpinnings of this improvement.

---

> > > > ### Comment · Reviewer_WB7G · 2024-11-29
> > > >
> > > > Thank you for the explanation and the additional results, which address most of my concerns. I will consider to update my scores given the additional results on Q1.

---

### Official Review · Reviewer_Cbf5 · 2024-11-05

**Soundness:** 2
**Presentation:** 3
**Contribution:** 2
**Rating:** 5
**Confidence:** 3

**Summary:**

The paper explores the use of Mamba-based multimodal LLMs to process long speech segments. The authors also apply DPO to enhance alignment during the instruction fine-tuning stage. Their experiments focus primarily on the speech summarization task, showing that the model can successfully process long speech.

**Strengths:**

The paper is clearly structured and represents a valuable exploration of using LLMs for long speech processing.

**Weaknesses:**

- Although the paper explores long speech processing using LLMs, the motivation is not sufficiently compelling. Notably, LLMs excel at summarization, and long speech summarization could be effectively handled by combining an ASR model with a strong LLM. While ASR may introduce some errors, LLMs are generally robust enough to manage this task. Therefore, long speech summarization may not be the most suitable task for evaluating LLMs in handling extended long speech inputs.
- The contribution is somewhat limited, as Mamba-based multimodal LLMs and DPO have both been explored in speech instruction tuning. This work primarily combines these two methods and tests them on the speech summarization task.
- The experiments are insufficiently comprehensive; relying only on speech summarization does not robustly support the model’s capability with long speech. Furthermore, the LLM used here is not particularly strong.

Overall, this paper lacks novelty and persuasiveness and would benefit from further development.

**Questions:**

I'm curious about the performance of advanced models (e.g., Whisper v3 and Llama 3) in building a cascade pipeline. Since this work aims to explore LLMs for speech processing, and LLMs are effective in handling long text and summarization tasks, how would these models compare?

---

> ### Author Response · Authors · 2024-11-25
>
> Thank you for the comprehensive feedback. We address each point raised below.
>
> # Question 1
> Although the paper explores long speech processing using LLMs, the motivation is not sufficiently compelling. Notably, LLMs excel at summarization, and long speech summarization could be effectively handled by combining an ASR model with a strong LLM. While ASR may introduce some errors, LLMs are generally robust enough to manage this task. Therefore, long speech summarization may not be the most suitable task for evaluating LLMs in handling extended long speech inputs.
>
> # Response
> We respectfully disagree with the reviewer's assessment for two key reasons:
> 1. The statement "long speech summarization may not be the most suitable task for evaluating LLMs in handling extended long speech inputs" seems to misunderstand our paper's objective. Speech summarization is not merely a test bed for evaluating LLMs - it is our primary task with significant real-world applications. Our goal is to improve speech summarization itself, rather than to evaluate LLMs' capabilities.
> 2. Regarding the suggestion of using ASR+LLM pipelines, prior work has already established that end-to-end models outperform cascaded approaches in speech summarization (Matsuura et al., 2023; Shang et al., 2024) by leveraging implicit acoustic features and minimizing error propagation, as mentioned in our Introduction. Our contribution lies in making these superior end-to-end approaches computationally feasible for longer inputs.
>
> # Question 2
> The contribution is somewhat limited, as Mamba-based multimodal LLMs and DPO have both been explored in speech instruction tuning. This work primarily combines these two methods and tests them on the speech summarization task.
>
> # Response
> While there have been several attempts at Mamba-based Vision LLMs, to the best of our knowledge, our work represents the first Mamba-based multimodal LLM architecture designed explicitly for speech summarization. More importantly, our primary contribution extends far beyond simply applying Mamba to speech summarization or combining it with DPO. The core technical innovation of our work lies in the querying-attention Mamba projector, a novel architectural component specifically designed to handle the unique challenges of long speech feature sequences. This projector introduces a new mechanism for efficiently condensing extended acoustic features into compact semantic tokens while maintaining linear computational complexity with input length.
>
> While we do utilize DPO for fine-tuning, this is secondary to our main architectural contribution. The efficiency gains and performance improvements demonstrated in our work stem from the novel design of our querying-attention mechanism and its specific optimization for speech summarization tasks, representing a significant advancement in efficient long-form speech processing.
>
> # Question 3
> The experiments are insufficiently comprehensive; relying only on speech summarization does not robustly support the model’s capability with long speech. Furthermore, the LLM used here is not particularly strong.
>
> # Response
> We respectfully disagree with these criticisms. First, as mentioned in our previous response, speech summarization is not merely a test case - it is our primary task with significant real-world applications. Our model's ability to handle long speech inputs is directly demonstrated through its performance on this challenging and practical task.
>
> Regarding the comment about LLM strength, we used the largest available open-source Mamba-based LLM for our implementation to ensure reproducibility and facilitate future research in this direction. While we acknowledge that stronger proprietary LLMs exist, we believe our choice represents a reasonable balance between model capability and research accessibility. The implementation of our architecture with stronger LLMs as they become available could be an interesting direction for future work.
>
> # Question 4
> I'm curious about the performance of advanced models (e.g., Whisper v3 and Llama 3) in building a cascade pipeline. Since this work aims to explore LLMs for speech processing, and LLMs are effective in handling long text and summarization tasks, how would these models compare?
>
> # Response
> We are currently conducting evaluations against a cascaded pipeline using Whisper v3 and Llama 3. While this pipeline may achieve higher performance given their larger model sizes (Llama 3 8B vs our 2.8B), our focus is on developing an efficient end-to-end architecture. Our benchmarks show that our Mamba-based approach achieves 2.4 seconds of inference time per sample compared to 41.7 seconds for the cascaded baseline. This 17x speedup stems from linear complexity scaling and the elimination of intermediate transcription. Thus, while larger models may achieve better absolute performance, SQuBa offers a valuable alternative for computationally-constrained applications.

---

### Author Response · Authors · 2024-11-29
**Revision with Human Evaluation**

We sincerely thank all reviewers for their thorough feedback and thoughtful suggestions. We have carefully considered all comments and made substantial revisions to strengthen the paper. Key additions and clarifications include:

1. Technical Clarifications
* Added explicit definition of h'(t) as time derivative dh(t)/dt
* Expanded justification for freezing the Whisper encoder, highlighting how it leverages pre-trained capabilities while maintaining computational efficiency
* Provided detailed explanation of why query length optimization is performed during alignment stage
* Clarified implementation details of cascaded baseline, including Whisper chunking and Mamba LLM configuration
* Added comprehensive explanation of our model's speed advantages, stemming from linear complexity scaling and elimination of intermediate transcription

2. New Experimental Results
* Added human evaluation results comparing our approach against baselines:
| Model vs. Model | win-rates (%:%)|
|----------------------|---------------------|
| SQuBa vs. Cascaded | **67.8** : 32.2|
| SQuBa vs. Kang & Roy | **92.6** : 7.4|
| Cascaded vs. Kang & Roy | **79.2** : 20.8|
* Conducted MOS evaluation of synthesized speech samples, achieving **4.16/5.0**

3. New Ablation Studies (Appendix D)
* D.1.1: Analysis of unidirectional Mamba's impact on model performance
* D.1.2: Comparative study of different downsampling methods
* D.1.3: Investigation of long speech alignment training effects

4. Enhanced Qualitative Analysis
* Added original transcripts alongside summaries
* Expanded qualitative results with additional examples

The full paper with these revisions and detailed experimental results is available. We hope these additions substantially strengthen the paper and address the key concerns raised in the reviews. We are additionally running coherency/consistency/fluency/relevance evaluations on the LLaMA-generated summaries of our dataset. We will report the results as soon as we get them.

---

> ### Author Response · Authors · 2024-12-02
> **Additional Evaluation on Synthetic Dataset**
>
> To address the concerns on the validation of our synthetic dataset and LLM-generated summary labels, we have conducted additional evaluations following the methodology of Jung et al. (ICASSP 2024) to assess the quality of our LLM-generated summaries. Using UniEval (EMNLP 2022), we evaluated both the ground truth summaries and our LLM-generated summaries across multiple criterions:
>
> | labels | Coherence | Consistency | Fluency | Relevance | Overall |
> |----------|------------------|------------------|-----------|-------------|-------------|
> | ground-truth | 85 | 82 | 89 | - | 84|
> | llm-generated | 91 | 88 | 94 | 82| 88 |
>
> Relevance denotes the relevance between the ground-truth summary and LLM-generated summary labels.
>
> These results provide comprehensive validation of our LLM-generated summaries across multiple evaluation dimensions. The generated summaries show strong consistency, coherence, and fluency performance, even exceeding the ground truth scores. The relevance score indicates that our generated summaries align strongly with the source content. Overall, the generated summaries achieve a strong aggregate score, comparing favorably with the ground truth score.

---

> > ### Comment · Reviewer_9c4a · 2024-12-02
> >
> > Please report human validation scores on these dimensions, as UniEval scores don't have any calibration here which makes it hard to assess LLM-generated label quality.

---

> > > ### Author Response · Authors · 2024-12-03
> > >
> > > We acknowledge that human evaluation scores would provide valuable additional validation of our LLM-generated labels. Our current use of UniEval follows the Jung et al. (ICASSP 2024) methodology, which employs both automatic metrics and human evaluation. While conducting comprehensive human evaluations requires significant time and resources that extend beyond the current review period, we agree this would strengthen our validation approach.
> > >
> > > While we agree that UniEval scores alone don't provide absolute calibration, they do enable relative comparison between ground truth and generated summaries, showing that our LLM-generated labels maintain comparable or superior quality across multiple dimensions.
> > >
> > > We are currently working on human evaluations following Jung et al.'s approach to provide additional validation and plan to include these results in the final version if possible. However, we believe our current automatic evaluation results, showing similar patterns to previously published work, provide meaningful preliminary evidence for the quality of our LLM-generated labels within the given time constraints.

---

> > > > ### Comment · Reviewer_9c4a · 2024-12-03
> > > >
> > > > I thank the authors for their engagement on this issue.
> > > >
> > > > It is not a concern restricted to UniEval (which is an automatic metric, and can be unreliable from the reviewer's own experience since the model was trained on LLM-generated summaries). The MOS scores in this case are similarly uncalibrated.
> > > >
> > > > Jung et. al. reports multiple metrics that measure similarity between source transcript and summary, and between existing and new reference summaries.
> > > >
> > > > Results from that paper are reported on both real and synthetic audio test sets, while, in this paper results are reported only on a synthetic audio test set. In my mind, that makes the burden of proof higher for this paper (meaning more evidence is needed).
> > > >
> > > > Based on all this, I have to concur with Reviewer TBjh that results on real audio datasets like SLUE-TED would be needed to be certain.

---

### Meta-Review · Area_Chair_BBJ3 · 2024-12-19

**Metareview:**

The paper introduces an end-to-end abstractive speech summarization method. It utilizes Whisper for speech encoding, a querying-attention Mamba projector to condense extended acoustic features into compact semantic tokens, which are subsequently fed into the Mamba Large Language Model (LLM). They further employ Direct Preference Optimization (DPO) fine-tuning to produce coherent summaries. Experiments on a TTS-synthesized speech summarization dataset demonstrate that this approach outperforms both a cascaded baseline and an end-to-end baseline.

Strengths

The model enables efficient speech summarization by combining pretrained Mamba and Q-former.
Empirical results are strong both in terms of performance and in terms of speed-up (efficiency)

Weaknesses

Human results (win rate) confirm that the proposed method is preferred over baselines (such as cascaded systems), but results on other suitable datasets such as How2 (no longer publicly available), SLUE_TED or AMI would make the results even stronger. The contribution is somewhat limited, as Mamba-based multimodal LLMs and DPO have both been explored in speech instruction tuning. This work primarily combines these two methods and tests them on the speech summarization task.

The additional results provided by the authors during the discussion lift the paper into the "can accept" category. since some reviewers increased their score, or expressed they would increase their scores. A full new review (confirming other clarifications have also been included) and results on more datasets would be needed to make this paper a clear accept given the somewhat limited novel (but good results and relevant task).

**Additional Comments On Reviewer Discussion:**

Good discussion during which the authors clarified several points and provided human evaluation results, which increased the value of the paper. However, results on a new dataset could not be presented during the rebuttal phase, which leaves some reviewers on the fence.

---

### Decision · Program_Chairs · 2025-01-22

Reject